# A high-resolution image time series of the Gornergletscher - Swiss Alps - derived from repeated UAV surveys

Lionel Benoit[1], Aurelie Gourdon[1,2], Raphaël Vallat[1], Inigo Irarrazaval[1], Mathieu Gravey[1], Benjamin Lehmann[1], Günther Prasicek[1,3], Dominik Gräff[4], Frederic Herman[1], Gregoire Mariethoz[1].

[1]Institute of Earth Surface Dynamics (IDYST), University of Lausanne, Lausanne, Switzerland.
[2]Ecole Normale Supérieure de Lyon, Département des Sciences de la Terre, Lyon, France.
[3]Department of Geography and Geology, University of Salzburg, Salzburg, Austria.
[4]Laboratory of Hydraulics, Hydrology and Glaciology (VAW), ETH Zürich, Zürich, Switzerland.

*Correspondence to*: Lionel Benoit (lionel.benoit@unil.ch)

**Abstract.** Modern drone technology provides an efficient means to monitor the response of alpine glaciers to climate warming. Here we present a new topographic dataset based on images collected during ten UAV surveys of the Gornergletscher glacial system (Switzerland) carried out approximately every two weeks throughout the summer of 2017. The final products, available at: https://doi.org/10.5281/zenodo.2630456 (Benoit et al, 2018), consist of a series of 10 cm resolution ortho-images, digital elevation models of the glacier surface, and Matching Maps that can be used to quantify ice surface displacements. Used on its own, this dataset allows mapping the glacier and monitoring surface velocities over the summer at a very high spatial resolution. Coupled with a classification or feature detection algorithm, it enables extracting structures such as surface drainage networks, debris or snow cover. The approach we present can be used in the future to gain insights into ice flow dynamics.

## 1 Introduction

Glacier ice flows by deformation and sliding in response to gravitational forces. As a glacier moves, internal pressure gradients and stresses create visible surface features such as glacial ogives and crevasses (Cuffey and Paterson, 2010). Furthermore, the surface of glaciers is also shaped by local weather conditions, which are responsible for the snow accumulation and ablation. Related processes generate distinct morphologies such as supra-glacial streams, ponds and lakes.

Glacier surface features evolve continuously, and these changes provide insights into the structure, internal dynamics and mass balance of the glacier. Important efforts have been made to monitor glacier surfaces, from early stakes measurements at the end of the 19[th] century (Chen and Funk, 1990) to present-day in-situ topographic surveys (Ramirez et al, 2001; Aizen et al, 2006; Dunse et al, 2012; Benoit et al, 2015) and remotely sensed data acquired from diverse platforms: ground-based devices (Gabbud et al, 2015; Piermattei at al, 2015), aircrafts (Baltsavias et al, 2001; Mertes et al, 2017) or satellites (Herman et al, 2011; Kääb et al, 2012; Dehecq et al, 2015; Berthier et al, 2016a). Recently, the development of Unmanned Aerial Vehicles (UAVs) has enabled glaciologists to carry out their own aerial surveys autonomously, rapidly, and at reasonable costs

(Whitehead et al, 2013; Immerzeel et al, 2014; Bhardwaj et al, 2016; Jouvet et al, 2017; Rossini et al, 2018). This technology is particularly attractive to map alpine glaciers whose limited size allows a satisfying coverage at a centimeter to decimeter spatial resolution.

Here we provide a homogenized and high-resolution remote sensing dataset covering about 10 km$^2$ of the ablation zone of the Gornergletscher glacial system (Valais, Switzerland, Fig 1). The raw images have been acquired by UAV flights carried out approximately every two weeks during the summer of 2017 (from May 29 to October 30). The dataset comprises 10 consecutive Digital Elevation Models (DEMs) and associated ortho-mosaics of the area of interest at a 10 cm resolution. It is therefore one of the most exhaustive surveys of the short-term surface evolution of a temperate glacier currently available. Geometrical coherence of the dataset is ensured through the application of a comprehensive photogrammetric processing (i.e. images are ortho-rectified and properly scaled). In addition, the orthomosaics are stackable thanks to a co-registration procedure. The dataset can therefore be seen as a high resolution time-lapse of the Gornergletscher ablation zone, combining spectral (orthomosaics) and geometrical (DEMs) information on the glacier surface. In addition to orthomosaics and DEMs that are snapshots of the area of interest, we also provide a product that we call Matching Maps (MMs) to achieve a temporal monitoring of the glacier. In practice, a Matching Map associates to each pixel of an orthomosaic (respectively a DEM) the corresponding pixel in the next orthomosaic recorded two weeks later. MMs can then be used to track the flow of ice over time, and in turn to quantify ice surface displacements.

Potential uses of this dataset are numerous. Single orthomosaics and DEMs can be used to map the surface of the glacier and to extract features of interest such as the surface drainage network (Yang and Smith, 2012; Rippin et al, 2015), debris or snow cover (Racoviteanu and Williams, 2012). Alternatively, the complete time series of orthomosaics and DEMs can be used for detection and quantification of changes at the surface of the glacier (Barrand et al, 2009; Berthier et al, 2016b; Fugazza et al, 2018). Finally, the time-lapse coupled with the Matching Maps is an interesting tool to monitor ice surface velocity and deformation (Ryan et al, 2015; Kraaijenbrink et al, 2016), and in turn ice flow dynamics at the glacier surface (Brun et al, 2018). The Matching Maps provide a quantification of the ice velocity at every location on the surface of the glacier, which can be used to calibrate or to validate ice flow models, especially for the Gornergletscher which was extensively used as a modeling benchmark (see for instance Werder and Funk, 2009; Riesen et al, 2010; Sugiyama et al, 2010; Werder et al, 2013).

## 2 Data acquisition

### 2.1 Study site

The Gornergletscher is located in the Valais Alps in southern Switzerland (Fig 1a). It is part of a glacier system involving five tributaries and ranges from 2200 m to 4634 m a.s.l. (Fig 1b). The ablation area, which is the main focus of this study, is a 4km long and relatively flat ice tongue (slope around 6 % i.e. 3.4°) that is deeply incised by meltwater channels and partially debris

covered (Fig 1c). This ablation area is preceded by a steeper part (south-west of the Monte Rosa Hütte, Fig 1c) characterized by the presence of numerous crevasses. The entire Gornergletscher system (i.e. the terminal tongue and its five tributaries) covers an area of almost 50 km$^2$ and its central flowline is 12 km long, making it one of the largest European glaciers (Sugiyama et al, 2010).

The Gornergletscher system has been widely studied since the 1970s due to its significant size, its accessibility, and because a glacier-dammed lake threatened the downstream Matter valley with glacier outburst floods (Sugiyama et al, 2010; Werder et al, 2009; Werder et al, 2013). The long history of glaciological studies in this area has shown that the mass balance of the Gornergletscher system was stable from the 1930s to the early 1980s, and significantly dropped since then (Huss et al, 2012). This can be associated with the rise of its Equilibrium Line Altitude (ELA) due to an increase of the local average yearly temperature. The ELA stands nowadays around 3300m according to studies carried out at the neighboring Findelgletscher (Sold et al, 2016).

In this context, the current dataset aims at complementing existing studies about the Gornergletscher system by documenting the behavior of its ablation zone during an entire summer, at a time when this glacial system is thought to be out of equilibrium with a clear trend toward glacial retreat. In particular, this dataset focuses on the monitoring of the glacier surface at high spatial and temporal resolution.

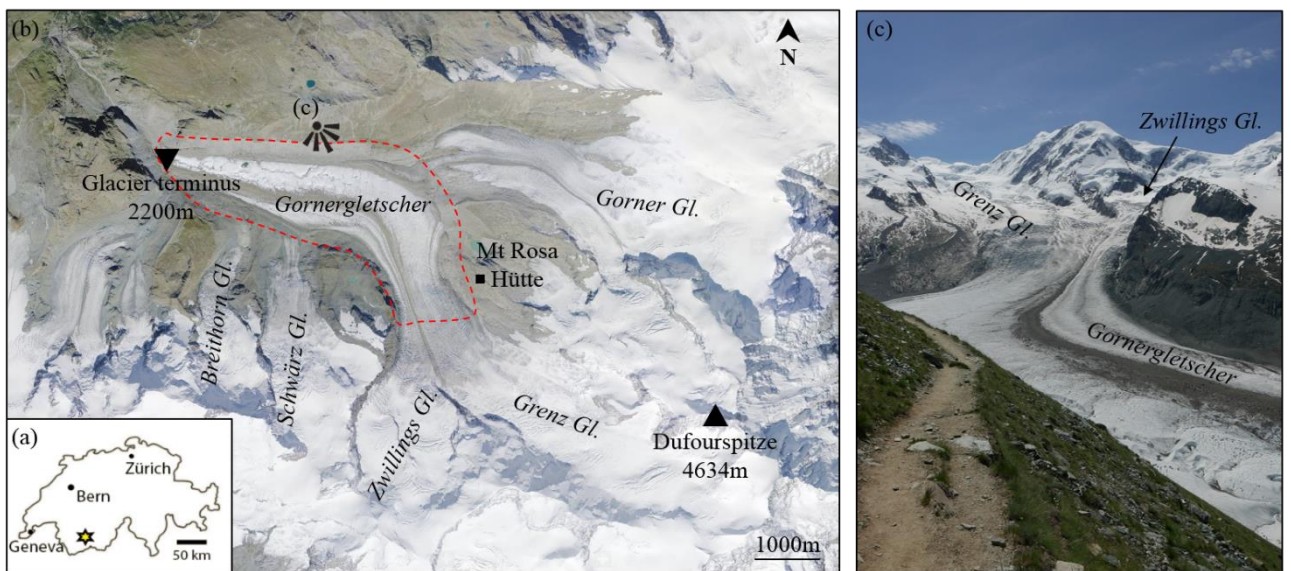

**Figure 1: The Gornergletscher system. (a) Situation map; (b) overview of the Gornergletscher system, dashed red line: area of interest; (c) picture of the glacier tongue.**

## 2.2 UAV surveys

The primary data are RGB images acquired by repeated UAV surveys over an area of 10 km$^2$. A fully autonomous fixed-wing UAV of type eBee from SenseFly, equipped with a 20 megapixels SenseFly S.O.D.A camera, has been used for image acquisition (Vallet et al, 2011). The camera was static within the UAV body (no Gimbal) and therefore all pictures are quasi-nadir (i.e. images are taken with an angle +/- 10° from the vertical line). For flight planning and UAV piloting, the eMotion3 software was used.

Raw images were acquired with a ground resolution ranging from 7.3 cm to 8.8 cm for the glaciated parts of the area of interest. For the requirements of photogrammetric processing, flight plans have been designed to allow for an overlap between images ranging from 70% to 85% in the flight direction, and from 60% to 70% in the cross-flight direction. These specifications have led to flight altitudes ranging from 300 m to 600 m above ground. The flight time was limited to about 30 min in field conditions. Thus, the coverage of the full area of interest required 4 to 8 separate flights per session, i.e. each day of acquisition (Table 1). Overall, 10 sessions have been conducted in 2017, from May 29 to October 30. The main features of these flights are summarized in Table 1:

| Date | Acquisition Time (CET) | # of flights | # of pictures |
|---|---|---|---|
| 2017/05/29 | 14:00 – 16:00 | 4 | 749 |
| 2017/06/09 | 12:30 – 15:30 | 8 | 935 |
| 2017/06/21 | 11:30 – 13:30 | 7 | 930 |
| 2017/06/27 | 11:30 – 14:00 | 5 | 1059 |
| 2017/07/13 | 12:30 – 14:00 | 4 | 830 |
| 2017/07/26 | 13:00 – 16:00 | 6 | 1125 |
| 2017/08/15 | 12:30 – 16:00 | 7 | 1121 |
| 2017/10/04 | 12:00 – 15:30 | 7 | 1107 |
| 2017/10/18 | 13:00 – 15:00 | 4 | 846 |
| 2017/10/30 | 13:00 – 15:30 | 6 | 1084 |

**Table 1: UAV flights carried out for raw glacier image acquisition.**

# 3 Data processing

## 3.1 Generation of co-registered orthomosaics and DEMs

Pictures have been processed separately for each acquisition date with the photogrammetric software pix4DMapper version 3.1 ((Vallet et al, 2011), Fig 2) using default parameters for nadir flights (see the processing reports for details about these parameters). The output resolution has been set to 10 cm/pixel in order to prescribe a constant resolution across all final products. During the photogrammetric processing, the raw pictures are first oriented by bundle adjustment, and then a DEM and an ortho-rectified image (orthomosaic) are generated for each day of interest. Since the only geolocation information included into the bundle adjustment procedure is the trajectory of the UAV derived from code-only GPS data, the initial geo-referencing of the orthomosaics and DEMs is limited to a few meters.

To improve the coherence of the co-referencing of the different sessions, all products are co-registered to the reference of the June 9 acquisition (Fig 2). To this end, the coordinates of several stable points of the landscape (16 to 70 among a set of 74, see Table 2 and Fig 4) are extracted from the bundle adjustment of June 9, and used as manual tie points for the bundle adjustments of the other dates. These stable points are mostly salient features of the bedrock or erratic boulders on the deglaciated banks of the glacier. The co-registration leads to orthomosaics and DEMs that are stackable. Therefore, in the final products, the bedrock remains stable between consecutive dates, while the glaciated parts move and deform. Consequently, if a time-lapse is created from the co-registered products, the glacier appears to flow while the surrounding landscape remains static. Fig. 2 summarizes the acquisition and processing chain used to derive the final products of the dataset.

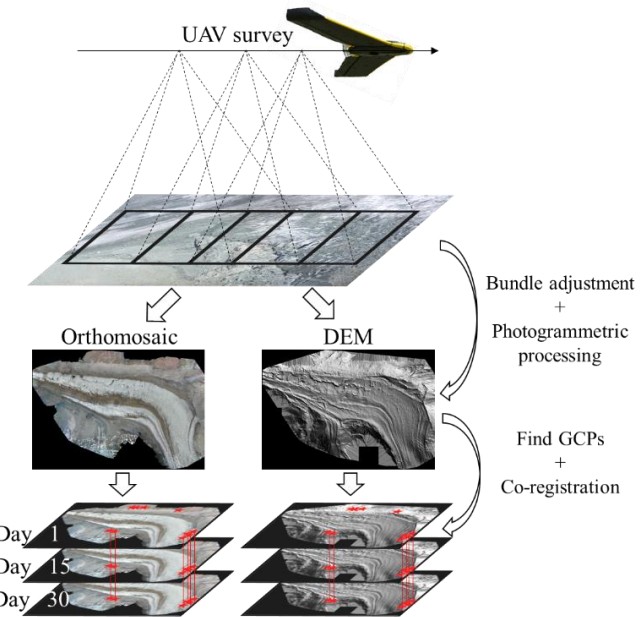

**Figure 2: Acquisition and processing chain used to derive the co-registered orthomosaics and DEMs**

After co-registration, all final products (orthomosaics and DEMs) are in the reference frame of the bundle adjustment of June 9 2017 (hereafter referred to as master bundle adjustment). This local reference frame is a realization of the WGS84 reference system (with Universal Transverse Mercator (UTM zone 32) projection) using code-only GPS data as input for referencing.

The absence of Ground Control Points (GCPs) and the use of consumer-grade GPS observations in the bundle adjustment procedure can result in meter-level geolocation errors and internal deformations of the master bundle adjustment (James and Robson, 2014; Gindraux et al., 2017). While the internal deformations of the local reference frame lead to relative measurement errors of small amplitude (on the order of 1/1000, see Sect. 4.1 for details), the geolocation errors related to the absence of GCPs can impair comparisons with other datasets covering the same geographic area. To improve absolute georeferencing and

to link our dataset with the Swiss national reference system, Table 2 provides the parameters of the affine transformation between the local reference frame of this dataset and the Swiss CH1903 – LV 03 reference system. This transformation has been estimated from 81 manual tie points identified in (1) the orthomosaic derived from the master bundle adjustment and (2) a 50 cm resolution orthomosaic of 2009 processed and georeferenced by the Swiss mapping agency SwissTopo.

| Translation Eastward (m) | Translation Northward (m) | Rotation (°) | Scale |
|---|---|---|---|
| 321800.8 | -5009609.3 | 1.1357 | 1.0004 |

**Table 2: Parameters of the affine transformation between the Swiss reference system (CH1903 – LV03) and the local reference frame defined by the master bundle adjustment of 2017 June 9. Note that no shear nor reflection is considered. Locations of the manual tie points used to estimate the transformation parameters are shown in Fig. 4.**

**3.2 Surface displacement tracking: generation of Matching Maps**

Consecutive co-registered orthomosaics enable to quantify horizontal displacements at the surface of the glacier. In the present dataset, this information about ice surface displacements is provided by the Matching Maps (MMs, Fig 3). In practice, a MM is an image that pairs the positions of similar ice patches at times $t$ and $t+dt$ ($dt$ being the time span between consecutive acquisitions) (Fig 2). The footprint of the MM is the overlap of the footprints of the orthomosaics at times $t$ and $t+dt$. MMs inherit the spatial resolution of the original orthomosaics (i.e. 10 cm) and can therefore be used to relate directly any pixel of

a given orthomosaic to its counterpart in the following orthomosaic (and therefore easily navigate within the whole dataset).

The MMs are obtained by image matching of pairs of orthomosaics. The orthomosaic at time $t$ is taken as a reference, and for each pixel of the reference, a 51 x 51 pixels (5.1 m x 5.1 m) patch is extracted and searched for in the orthomosaic corresponding to the next session (time $t+dt$). To speed up the processing and avoid wrong matches with very distant patches,

the homolog patch at time $t+dt$ is searched for in a neighborhood with a 200 pixels (20m) radius centered on the position of the original patch at time $t$ whose size has been established based on prior knowledge about the approximate surface velocity

of the Gornergletscher. The criterion used to evaluate the similarity between both patches is the mean absolute error (MAE) between pixels computed on grayscale images (Liu and Zaccarin, 1993; Chuang et al, 2015). The MAE has been selected as similarity score because it is fast to compute, especially on large images using convolutions. Its disadvantage is the sensitivity to illumination differences between consecutive orthoimages. However, in practice, no adverse effects have been observed, mostly because the images were acquired roughly at the same time of the day (between 11:30 and 16:00), and because the orthoimages used to generate the matching maps are always separated by less than one month, which mitigates the illumination differences. The patch of the image $t+dt$ leading to the lowest MAE with the original patch at time $t$ is then considered the counterpart of the original patch. Finally, the displacements (in pixels) between the two patches along the East-West and the North-South directions are recorded into the MM. This operation is repeated for all possible patches in the reference orthomosaic. The MMs have been calculated using an open source utility called MatchingMapMaker developed as part of this project, and made available along with the dataset (See Sect. 5.2 for code availability). The MatchingMapMaker tool has been designed to account for the specificities of UAV-based orthoimages, and in particular their very high resolution. To ensure the reliability of this utility, MMs have been benchmarked against horizontal displacement maps calculated using well-established image correlation algorithms, namely Imcorr (Scherler et al., 2008) and CIAS (Heid and Kääb, 2012). The results of this benchmark (see Sect 5.2) show a very good agreement between horizontal displacement maps derived from MatchingMapMaker, Imcorr and CIAS.

The raw MMs can be noisy due to the presence of outliers in the pattern matching procedure (speckled areas in the raw displacement maps in Fig 3). These outliers originate from the dissimilarity between subsequent orthomosaics, due to, for example, changing shadows or changes at the glacier surface (snowfall, snow or ice melting, etc.). To mitigate the impact of these outliers, we first locate them, then we mask the impacted areas (pink areas in Fig. 3), and finally we interpolate the remaining reliable displacements to fill the gaps generated by the mask. To limit the processing time, a simplistic outlier detection method based on signal processing has been preferred over more sophisticated approaches based on glacier physics (Maksymiuk et al, 2016). Unreliable areas in the raw MMs are assumed to be aggregates of pixels with spatially incoherent displacement values embedded in a matrix of displacements that vary smoothly in space (i.e. the reliable displacements). The borders of unreliable areas are detected as locations with strong spatial displacement gradients, with a detection threshold set to 15 cm of horizontal deformation per day. A mask of reliability is then created by setting the areas with strong gradient to 0 and the remaining of the mask image to 1. The outlier areas (i.e. small aggregates of unreliable values) are then filtered out by applying the opening operator of mathematical morphology to this mask with a structuring element of size 50 x 50 pixels. This operation leads to switch the value of the mask from 1 to 0 for all aggregates of pixels smaller than 50 x 50 pixels. Hence, we obtain a mask with 1 at locations with reliable displacements and 0 where the measured displacements are considered as outliers. Finally, the values of the MM at masked locations are interpolated from the non-masked measurements using a bilinear interpolation. The selected procedure is iterative. At each iteration, it attributes to the masked values the mean of the reliable values in a 500 pixel neighborhood in the East-West and North-South directions. The values that remain masked after 10

iterations are considered as too far from the informed areas to be filled and are set to -99 to denote no data. Fig. 3 summarizes how MMs are derived from pairs of consecutive co-registered orthomosaics and filtered to remove outliers.

Because MMs pair the positions of similar ice patches between consecutive orthomosaics, they can be used to derive maps of the horizontal displacements occurring at the surface of the glacier. To this end, displacements from the masked MMs are converted in m/day and resampled at a 5m resolution to remove the dependence between neighboring locations that is introduced during the image matching procedure. Horizontal displacement maps at 5 m resolution are provided in addition to the full resolution MMs in order to facilitate the use of the present dataset in the context of ice flow studies.

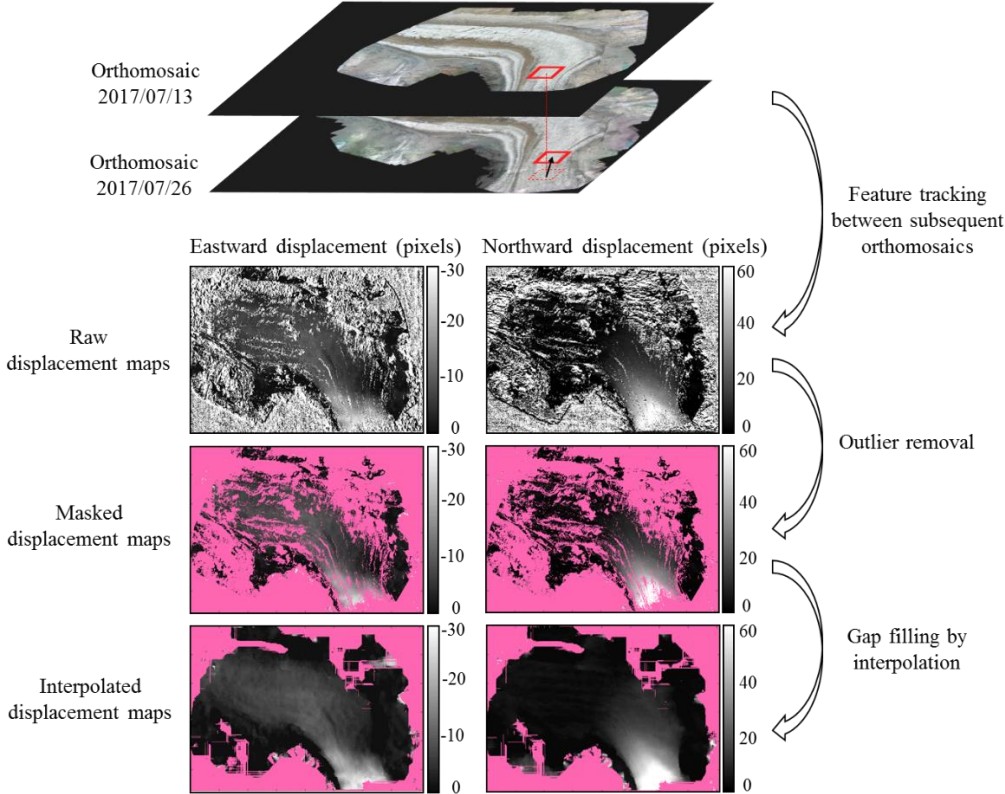

**Figure 3: Processing chain used to compute a matching map between two subsequent orthomosaics. The procedure is illustrated for the 2017/07/13 – 2017/07/26 period. In displacement maps, masked areas are displayed in pink.**

# 4 Quality assessment

## 4.1 Bundle adjustment and co-registration

A first validation of this dataset can be done by checking the relative orientation of the cameras during the bundle adjustment, as well as the co-registration of orthomosaics and DEMs. Processing reports detailing the quality of the bundle adjustment for
each session are available along with the dataset (see Sect. 5.1).

Table 2 displays three indices summarizing the quality of both bundle adjustment steps. First, the mean reprojection error (in pixels) quantifies the mismatch in the raw images between the observed and the modelled position of tie points used during the relative orientation step. The sub-pixel level of errors (Table 2, column 2) ensures that the orientations of the camera are
reliable. Next, the co-registration step is assessed by the mean Root-Mean-Square (RMS) error of manual tie point coordinates. This statistic measures the stability of manual tie point coordinates between different bundle adjustments. Under ideal conditions, the value of the mean RMS error on manual tie points should be close to the ground pixel resolution of the raw images (i.e. 7.3 cm to 8.8 cm) because an operator is able to identify points of interest with a pixel level precision. The slightly higher values obtained in the present case (9 cm to 21 cm, Table 2, column 3) can be due to the difficulty of precisely identifying
manual tie points under changing environmental conditions (e.g. sunlight exposition or snow cover). The errors in manual tie point identification degrade the mean RMS error, but they are expected to have a mild impact on the co-registration itself because they are not correlated and tend to compensate each other. Note that late in the season (i.e. for the last acquisition on October 30[th]) it became difficult to identify manual tie points due to strong shadows, hence the small number of manual tie points at that time.

| date | Relative orientation: mean reprojection error (pix) | Co-registration: Mean RMS error (m) | Co-registration: # manual tie points |
|---|---|---|---|
| 2017/05/29 | 0.138 | 0.210 | 66 |
| 2017/06/09 | 0.136 | Reference | Reference |
| 2017/06/21 | 0.123 | 0.189 | 66 |
| 2017/06/27 | 0.146 | 0.193 | 68 |
| 2017/07/13 | 0.120 | 0.107 | 63 |
| 2017/07/26 | 0.117 | 0.206 | 70 |
| 2017/08/15 | 0.118 | 0.175 | 69 |
| 2017/10/04 | 0.125 | 0.122 | 38 |
| 2017/10/18 | 0.127 | 0.146 | 43 |
| 2017/10/30 | 0.125 | 0.092 | 16 |


**Table 2: Quality assessment of the bundle adjustment procedure.**

Another important validation consists of assessing possible internal deformations within the local reference frame of the dataset. Fig. 4a displays the residuals of the co-registration of the master bundle adjustment on a georeferenced orthoimage, which are a proxy for the internal deformations of the master bundle adjustment. The results show that the internal deformations have a meter-level amplitude (mean deformation = 1.07 m, max deformation = 2.83m) and are smoothly spread over the area of interest due to the bundle adjustment procedure, which tends to distribute errors over space. It follows that, considering the extent of the study area (few square kilometers), the relative error induced by the internal deformations of the local reference frame is on the order of 1/1000. Thanks to the co-registration procedure, the internal deformations of all orthomosaics and DEMs are similar to the ones of the local reference frame defined by the master adjustment. When measuring changes at the surface of the glacier from the present dataset, the error related to internal deformations is therefore on the order of one per mille of the measured distances. This results in relatively small absolute errors because the changes at the ice surface of the Gornergletscher are of moderate amplitude (e.g. ice ablation reaches few cm/day, and ice flows at less than 1m/day in the ablation zone). For instance, in case of horizontal velocity measurements, the order of magnitude of glacier displacement between two acquisition dates is 30 cm/day x 14 days = 4.2 m. It follows that the error in velocity due to internal deformations is 4.2 m x 1/1000 /14 days=0.3 mm/day, which is very small in comparison with the amplitude of the ice surface velocity itself.

**4.2 Orthomosaics and DEMs**

In addition to the bundle adjustment, we also validate the final products of the photogrammetric processing (Fig 4a), i.e. the co-registered orthomosaics and DEMs. To this end, individual orthomosaics and DEMs have first been visually checked to track the presence of artifacts. A careful examination of all products shows that the glaciated parts (Fig 4b) as well as the neighboring ice-free areas (Fig 4c) are well reconstructed in both orthomosaics and DEMs. On the edges of the area of interest artifacts can be present due to the low number of overlapping images in these areas (see the processing reports to identify them). This leads to unreliable photogrammetric reconstructions and in particular shear lines (Fig 4d). Despite these relatively minor artifacts restricted to the edges of the surveyed area, all glaciated parts and nearby unglaciated margins are satisfyingly reconstructed in both orthomosaics and DEMs.

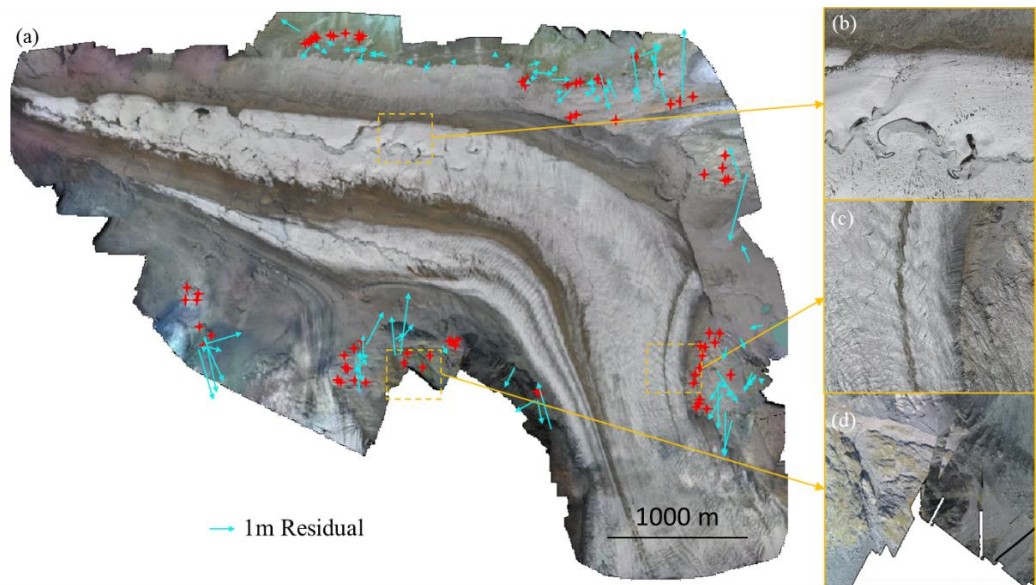

**Figure 4: Quality assessment of the orthomosaics. (a) Overview of one orthomosaic (2017/08/15). Red stars: manual tie points used for co-registration. Blue arrows: residuals after co-registration of the master bundle adjustment on a 50 cm-resolution orthoimage acquired in 2009. The affine transformation described in Table 2 has been used for co-registration. (b) - (c): Examples of areas where the photogrammetric processing worked properly. (d): Example of area on the boundary of the domain where the photogrammetric processing produced artifacts (mostly shear lines).**

## 4.3 Matching Maps

In addition to the visual inspection of individual photogrammetric products, we also assess the quality of the co-registration procedure by quantifying in the MMs the stability of several areas that are most likely static, as well as the observed spatial patterns of glacier surface velocity. We select several validation locations on and off the glacier (Fig 5) and compute their horizontal velocity by dividing the displacements recorded in the MMs by the time elapsed between the acquisitions. Note that in Fig. 5 the velocity is averaged over 10 x 10 m$^2$ areas, corresponding to 10000 single measurement points, centered on the validation points.

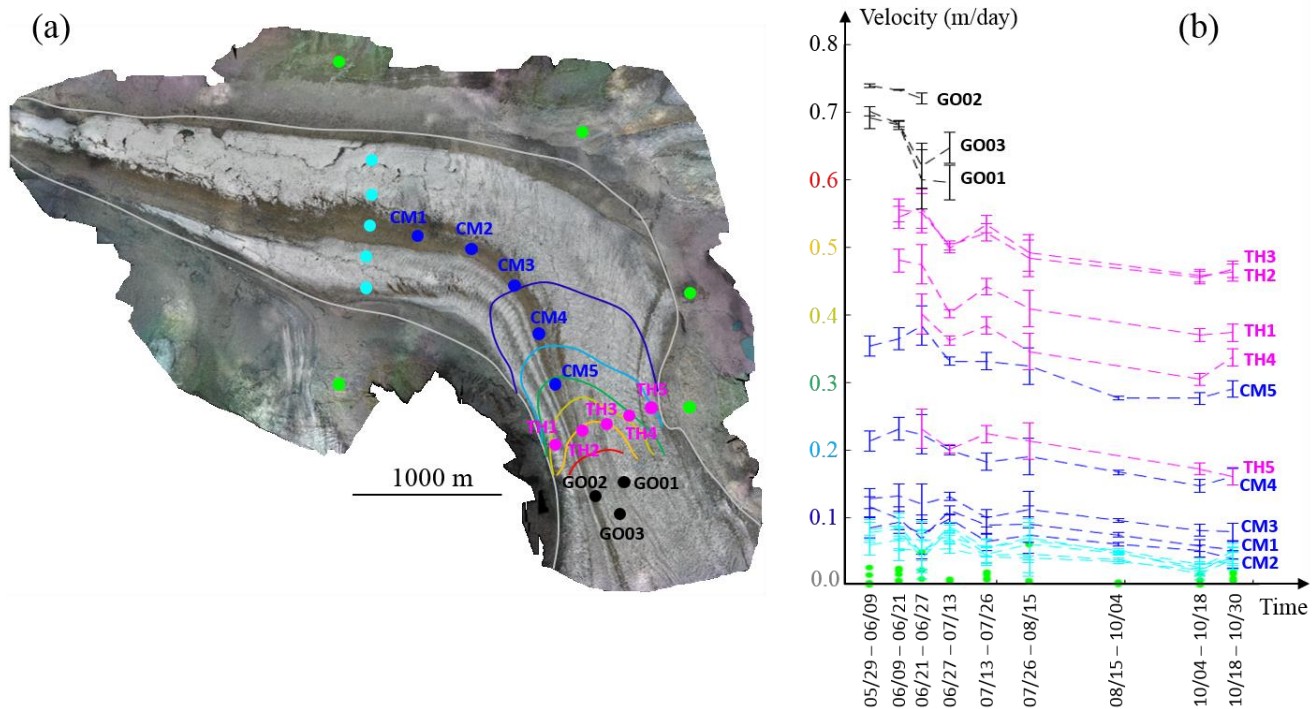

**Figure 5: Quality assessment of the Matching Maps. (a) Locations of the validation points. The contour lines represent the horizontal surface velocity derived from the MM related to the period 07/13 – 07/26. (b) Observed horizontal surface velocities at validation locations. Error bars show 1σ errors. The errors reported for UAV-based velocities are empirical errors and are equal**

5 **to the quadratic mean of velocities recorded at ice-free locations (green dots in (b)). The errors reported for GNSS-based velocities (i.e. at locations GO01, GO02 and GO03) are theoretical errors accounting for the uncertainty induced by the tilt of the support of GNSS receivers over time due to glacier movement.**

Fig. 5b displays the observed horizontal velocities in the domain for summer 2017. In case of perfect photogrammetric

10 processing, co-registration, and feature tracking, the apparent velocity of the ice-free areas (in green on Fig 5) should be zero. While it is not exactly the case due to inherent processing errors and measurement noise, the mean velocity is very low (1.2 cm/day on average over the 5 ice-free validation locations) which reflects an appropriate processing. The observed patterns of glacier surface velocity are also in accordance with typical patterns of ice flow, such as velocities decreasing from the center of the glacier towards the edges (compare e.g. the velocity in TH3 and TH5) and higher velocities at steep parts of

15 Grenzgletscher than on the flat tongue of Gornergletscher (compare e.g. TH2 and CM5 to CM1). Finally, the velocities derived from UAV correspond to independent data collected by differential GNSS measurements a few hundred meters upstream of the area of interest (points GO01, GO02 and GO03 in Fig 5). The higher velocity measured at the locations monitored by GNSS (points GO01-G003) compared to the downstream locations monitored by UAV (points TH1-Th3) is coherent with the increase of glacier velocity at the steeper upstream part of Grenzgletscher (approx. 13.5 % at GO02 compared to 7.5 % at

TH2). Finally, the trend of deceleration over the course of the summer recorded by GNSS is in good agreement with the UAV-based velocities throughout the glacier.

## 5 Data and code availability

### 5.1 Structure and availability of the dataset

All the data presented in this dataset are available in the following repository (Rep): https://zenodo.org/record/1487862, with the following DOI: https://doi.org/10.5281/zenodo.2630456 (Benoit et al, 2018).

The results of the photogrammetric processing, i.e. the orthomosaics and the DEMs, are available in the compressed folder Rep\Photogrammetric_Products.zip. Within this folder, the products are grouped in sub-folders by acquisition date using the following standard: 2017_mm_dd with 'mm' the month and 'dd' the day of acquisition. Finally, these sub-folders contain the following files:

- 2017_mm_dd_orthomosaic.tiff: Contains the orthomosaic.

- 2017_mm_dd_dem.tiff: Contains the DEM.

- 2017_mm_dd_report.pdf: Contains the processing report (generated by Pix4D Mapper) that summarizes the quality of the photogrammetric processing for the date of interest.

The Matching Maps are stored in the compressed folder Rep\Matching_Maps.zip. Within this folder, full resolution MMs are stored in the \Full_resolution_Matching_Maps sub-folder. In this folder, individual maps are grouped in sub-folders named according to the acquisition date of the pair of subsequent orthomosaics used to generate the Matching Map: 2017_mm_dd_2017_nn_ee with 'mm' (resp. 'dd') and 'nn' (resp. 'ee') the acquisition months (resp. days). These sub-folders contain the following files:

- 2017_mm_dd_2017_nn_ee_disp_Eastward: Contains the Matching Map of Eastward displacements.

- 2017_mm_dd_2017_nn_ee_disp_Northward: Contains the Matching Map of Northward displacements.

- 2017_mm_dd_2017_nn_ee__disp_mask: Contains the mask of reliable displacements after filtering: 1 if the location corresponds to a reliable displacement, 0 otherwise.

In addition to the full resolution MMs, displacement maps at 5 m resolution are stored in the \ Final_Displacement_Maps sub-folder. Note that in contrast to the MMs, the displacement maps are in m/days. Displacement maps follow the same file nomenclature as MMs, except the _Res5m suffix that allows to distinguish displacement maps from MMs.

**5.2 Code availability**

The photogrammetric processing has been carried out using the proprietary software Pix4D Mapper, commercially available at: https://pix4d.com/ (last access 2018/11/16).

The Matching Maps have been computed using Matlab routines written by Mathieu Gravey. The related utilities are freely
available on the following repository: https://github.com/GAIA-UNIL/MatchingMapMaker. The sub-repository Benchmarking_tests contains the results of benchmarking tests aiming at comparing the displacement maps computed by the MatchingMapMaker utility (i.e. MMs) with displacement maps computed by well-established glacier surface tracking algorithms, namely Imcorr (https://nsidc.org/data/velmap/imcorr.html) and CIAS (https://www.mn.uio.no/geo/english/research/projects/icemass/cias/). The sub-repository Similarity_score_tests contains the
results of tests assessing the sensitivity of the MatchingMapMaker output to the similarity score used to define patch matches.

**6 Conclusion**

The present dataset compiles ten UAV surveys of the Gornergletscher carried out during summer 2017. Photogrammetric processing leads to a set of 10 cm resolution orthomosaics, DEMs and glacier displacement maps for each acquisition date. This dataset can be used for change detection and quantification at the glacier surface, and in particular to investigate glacier
surface dynamics at high temporal and spatial resolution.

**Author contributions**

AG, FH and GM designed the experiment.

AG, RV, II, BL, GP and LB carried out the acquisitions.

AG, RV and LB performed the photogrammetric processing.
MG, LB and AG computed the Matching maps.

DG recorded differential GNSS data used for validation.

LB wrote the manuscript with inputs from all authors.

**Competing interests**

The authors declare no competing interests.

## Acknowledgments

The authors are grateful to Philippe Limpach from ETH Zürich who processed the GNSS data.

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
