# Peer review of "A high-resolution image time series of the Gornergletscher - Swiss Alps - derived from repeated UAV surveys"

_Earth System Science Data, 2018_

## Referee Comment (RC1) · Anonymous Referee #1 · 11 Jan 2019

The authors present an elevation and velocity dataset over the lower part of the Gorner glacier in Switzerland. These products are generated from several field campagnes during the summer of 2017. Making this dataset available to a large community is of interest, as the evolution of surface features can be observed in great detail. It can therefore be of interest to geomorphology and glaciology.

In general the article is well written, and the processing steps of the workflow are described sufficiently. However, the motivation for several processing steps are lacking explanation, and implications of such decisions not mentioned. Two concerns are currently present in the manuscript, which need to be addressed, in order to have its full

potential for other users.

The first concern is based on the relative geo-referencing, the authors have chosen not to use real ground control points. While they have chosen one common bundle-adjustment as master, to "stitch" other control against it. It is common in photogrammetry to distribute GCP's and have them placed especially at the outer ends. This reduces "banana" bending, caused by imperfections in the lens model. However, this is not done in this study, thus one can assume such effects are here at hand as well. I am well aware of the logistics within such terrain, thus I am not asking to do this procedure. Nevertheless, I propose the authors put a bit more emphasis into describing the potential errors associated with this effect/shortcoming. I hope if this is done rightfully, it will reduce the mistake of over interpretation by other users of this data.

My second concern is focused around the matching maps, which are less standard products, and therefore implementation details need to be discussed more. Although not standard, there do exist best-practices, and because the authors deviate from this I highlight some steps which may need more clarification, or adjustments, in order to improve the resulting matching map product or get a better understanding why certain steps are taken.

- The similarity score is "maximum absolute error" on grayscale images. This is a peculiar choice as throughout the season and throughout the day the sun has changing illumination directions. Though this similarity measure is very sensitive towards such effects. Commonly, the normalized cross correlation is used, and the images are either pre-filtered with a high-pass-filter [Fahnestock 2016] or a Wallis-filter [Dehecq 2018]. I expect this will improve the results considerable. Also the use of orientation-correlation [Heid 2012] or COSI-CORR [leprince 2007] might be a more robust procedure. - The matching maps seem to have an integer displacement, and no sub pixel localization is applied. While these procedures are able to increase the precision considerably. This will result in highly precise maps, where strain rates and other derivatives can be extracted more precisely. Therefore, I wonder why this is not done. - The spacing of velocity product is at pixel level, though in products like GoLIVE for example, the spacing of the grid is as large as the template size (in this case 300 meters). Similar processing in SAR speckle tracking is done with only 50% overlap [van wychen 2018]. This is done to have independent measurements, but now there will be a large smearing effect. It is possible to get to the resolution of a pixel, when pyramidal matching is applied, up to a point where optical flow can to be implemented.

p = page; l = line; -> = consider changing to;

minor comments:

p1 l1 high-frequency -> inter-seasonal

p1 l1 image -> elevation and velocity

p1 l13 new dataset -> new topographic dataset

p1 l13 intensive: subjective text, I am aware this is hard work but not necessary information for the abstract

p1 l14 summer 2017 -> summer of 2017

p1 l17 displacements and velocity, choose one as these are two words for the same thing

p2 l7 summer 2017 -> summer of 2017

p2 l8 -> "10 consecutive DEMs and associated ortho-mosaics"; you first need to make a DEM in order to be able to make a mosaic

p2 l9 surface evolution

p2 l17 displacements and velocities; choose one

p2 l23 "ice flow dynamics at the surface" I think the emergence velocity by [Brun 2018] is also a good example of a process. Furthermore, mentioning the use of multitemporal DEM's as in [Wang , Berthier ] might be worth it as well.

p3 l1 "6%" maybe also specify in degrees

p3 l4 "mainstream" -> central flowline?

p3 l19 maybe include the lake location, which drains, and also include the ELA?

p5 l3 pix4D, please give the version

p5 l7 swap orthomosaic and DEMs

p5 l9 UTM_zone32 -> Universal Transverse Mercator (UTM zone 32)

p5 l12 I assume all flights are nadir, or did the UAV also took oblique imagery. This is of interest, as it enhance the separation between internal parameters [james 2014].

p6 l1 "GCPs" is maybe not the correct term, as they are not real ground control, hence (manual) tie-points might be more correct.

p6 l10 51x51 pixels , also include the metric size, that is roughly 5 meters right?

p6 l12 2000 pixels, see former comment

p7 l10 "locations with strong spatial gradients", this is not a very effective post-processing step. It is very local and isolated inliers will also be removed. More advanced post-processing techniques are possible [maksymiuk 2016], but I am not asking to do this, just so you are aware of such studies.

p7 l16 "are interpolated", which type/method?

p7 l16 "reliable measurements", not correct wording, there is no real testing, thus reliable is misplaced here.

p8 l1 the east and northward components do not seem to be the same....? also, why is the displacement given in pixel and not in a metric scale?

p8 l14 There seem to be multiple flights per campaign. Because a fixed wing is used, the landing must have had an impact. Hence, internal camera parameters might have

been different between different flights. Thus, are these groups of data also separated in the camera optimization?

p9 l3 are due -> can be

p10 l11 "smooth out local variability", why is this done? and why is the mean taken, and not more robust measures like the median?

p11 l12 why are single points used as validation, it might be good to look at stable terrain as well. The Randolph glacier mask can exclude glacial terrain, the rest can be used to create a histogram of displacements.

p11 l12 "since the area of interest", is only the icefall of interest? Are other features on the glacier tongue of interest as well (meander evolution of supra glacial streams, emergence velocity ....)?

p13 l4 "for almost any glacier mapping task", not specific and does not hold either, please rephrase.

[berthier 2016] Decadal region-wide and glacier-wide mass balances derived from multi-temporal ASTER satellite digital elevation models. Validation over the Mont-Blanc area

[brun 2018] Ice cliff contribution to the tongue-wide ablation of Changri Nup Glacier, Nepal, central Himalaya

[dehecq 2018] Twenty-first century glacier slowdown driven by mass loss in High Mountain Asia

[fahnestock 2016] Rapid large-area mapping of ice flow using Landsat 8

[heid 2012] Evaluation of existing image matching methods for deriving glacier surface displacements globally from optical satellite imagery

[james 2014] Mitigating systematic error in topographic models derived from UAVand

ground-based image networks

[leprince 2007] Co-registration of optically sensed images and correlation (COSI-Corr): An operational methodology for ground deformation measurements

[maksymiuk 2016] Velocity estimation of glaciers with physically-based spatial regularization—Experiments using satellite SAR intensity images

[van wychen 2018] Surface Velocities of Glaciers in Western Canada from Speckle-Tracking of ALOS PALSAR and RADARSAT-2 data

[wang 2015] Modeling glacier elevation change from DEM time series.
* * *

---

## Referee Comment (RC2) · Anonymous Referee #2 · 14 Jan 2019

The manuscript "A high-frequency and high-resolution image time series of the Gornergletscher - Swiss Alps - derived from repeated UAV surveys" presents a series of UAV derived datasets for the Gorner glacier in Switzerland collected at $\sim$two weekly interval over the summer of 2017. Datasets include, DEM's and "matching maps" (velocity fields) for each date at high (centimetre) spatial resolution, over a relatively large area of the glaciers ablation zone. As such the dataset exploits both the spatial and temporal benefits of UAV data collection and could thus be useful to others in glaciology, and more broadly within geomorphology and the earth sciences. In general the article is well written, datasets are properly documented and the processing workflow is generally adequately described. However, some issues with what I would consider

non-standard nomenclature and processing methods, e.g. the matching maps makes the article somewhat difficult to follow.

Despite these positive aspects I believe there is a fundamental issue in the dataset that limits its suitability for use and widespread dissemination, and therefore I unfortunately cannot recommend publication until this has been remedied. These issues are related to the lack of ground control, which I will expand on below.

There are three typical methods used for positional/measurement accuracy of UAV and structure from motion (SfM) photogrammetric surveys, and a number of examples in glaciologic applications (Bhardwaj et al., 2016). 1) in scene distance measurements – where features of known length are distributed across the scene, e.g. 1m length rulers, or distances between features that are manually measured. This can provide accurate distances within a scene but not absolute positioning in space, vertical distances are also often prone to error with this method. 2) In scene ground control points – where targets are installed and surveyed with differential GNSS (cm accuracy) and used to force the SfM model into real world coordinates. This is probably the most widely used method, e.g. (Hugenholtz et al., 2013; Immerzeel et al., 2014; Westoby et al., 2012; Whitehead et al., 2013; Wigmore and Mark, 2017) and when ground control is well distributed provides the most accurate results. 3) positioning of the UAV, generally less accurate than in scene ground control and heavily dependent on positional accuracy of the UAV platform. The gold standard is dual phase L1/L2 differential PPK/RTK GNSS positioning combined with high resolution IMU measurements (∼cm accuracy of UAV positon), poorest quality is using consumer grade GPS (∼3m horizontal, ∼10m vertical error).

In this case it appears the authors used the later method relying on consumer grade GPS positions of the UAV to generate the 9 June orthomosaic and DEM, all other datasets are then coregistered to this 'master' dataset using stable bedrock features. What this means is that the stacks themselves are co-registered accurately and we can therefore observe changes between images pairs fairly precisely. However, the

master 9 June dataset is likely to be so inaccurate positionally that any measurements made from these data are likely to be highly inaccurate. A number of papers of have investigated the errors associated with using no/insufficient ground control in conjunction with low accuracy platform positioning (Gindraux et al., 2017; Hugenholtz et al., 2016; James et al., 2017; James and Robson, 2014, 2012; Tonkin et al., 2014; Tonkin and Midgley, 2016). Furthermore these products are highly likely to result in significant 'doming' or 'fishbowling' where the edges of the SfM survey area curve upwards or downwards (James and Robson, 2014). Both horizontal and vertical errors are highly likely. These publications suggest that these combined positional errors are likely on the order of 1-10m in the horizontal and vertical which means any slope, distance, velocity, elevation, etc measurements made from the DEM are probably highly inaccurate. Given that measured amounts of glacier change reported in this article are less than the magnitude of the potential error, I do not believe this is acceptable. Furthermore there is no external validation of the datasets accuracy – using either additional ground control targets, LiDAR or other datasets. As such I don't believe this data should be published in its current form. Below I have outlined a few potential methods for remedying this issue.

1) Install ground control targets in the survey area and conduct another survey of the glacier, then use this as the master dataset to co-register the other datasets to, using stable/no change locations. This is that was done in the (Immerzeel et al., 2014; Kraaijenbrink et al., 2016) articles that the authors use as justification for the co-registration method they employed. Obviously installing ground control targets on glaciers and over a large area is extremely difficult and often not possible given access, safety and logisitical challenges. This is one of the major limitations of collecting research grade UAV datasets over challenging environments. Because of these limitations we are often forced to live with poorly distributed or inadequate numbers of ground targets. In the glacier context this means ground control is often restricted to installations along the moraine edges, but even this is much better than none. And, would in my opinion provide an acceptable product in line with other UAV/glacier publications. Furthermore,

the authors did use some 'on-ice' GNSS data to assess the accuracy of the velocity measurements - these stations could be used in conjunction with moraine ground control targets alongside perhaps two more on ice targets - e.g. one near terminus, one near middle, limiting labour/time investment. 2) Fly a UAV equipped with L1 or L1/L2 differential GNSS and use an RTK or PPK positional solution to derive a more accurate master data set. Then use this dataset to co-register the other dates. An example of this method as applied to glaciers is currently under review in the cryosphere discussions: Chudley et al., 2018. 3) If there is a high quality DEM and imagery data from other sources e.g. LiDAR/airborne imagery/high resolution satellite data, then authors could extract ground control positions from these data and co-register the UAV datasets to this. If available the raw point clouds could also be co-registered using cloud compare or similar software. This method is likely to be the least accurate as it depends on the positional accuracy of the base data used. However, it is the easiest to implement and would provide an acceptable level of accuracy in my opinion, and given the challenges of 1) and 2) above may be the most feasible.

Given the significant issue outlined above I have omitted a detailed line by line review of grammar/spelling etc., but am happy to do so once my main concern has been dealt with. Finally, I have read the other posted review for this paper and agree with their suggestions regarding the matching maps. Further description of how these were derived is needed as they are non-standard, and ideally more widely used methods for velocity field derivation should be applied – e.g. COSI-CORR as this would be 1) easier to follow, 2) likely to be more useful to other researchers.

Additional points: To prevent the use of incorrect data by others the authors should clip out obviously erroneous regions in the DEM and orthomosaic, i.e. along the edges of the survey area. Ideally all datasets could be clipped to a common extent boundary and raster grid for ease of use, though this is less important.

Bhardwaj, A., Sam, L., Akanksha, Martín-Torres, F.J., Kumar, R., 2016. UAVs as remote sensing platform in glaciology: Present applications and future prospects. Remote Sens. Environ. 175, 196–204. doi:10.1016/j.rse.2015.12.029

Chudley, T. R., Christoffersen, P., Doyle, S. H., Abellan, A., and Snooke, N.: High accuracy UAV photogrammetry of ice sheet dynamics with no ground control, The Cryosphere Discuss., https://doi.org/10.5194/tc-2018-256, in review, 2018.

Gindraux, S., Boesch, R., Farinotti, D., 2017. Accuracy assessment of digital surface models from Unmanned Aerial Vehicles' imagery on glaciers. Remote Sens. doi:10.3390/rs9020186

Hugenholtz, C., Brown, O., Walker, J., Barchyn, T., Nesbit, P., Kucharczyk, M., Myshak, S., 2016. Spatial accuracy of UAV-derived orthoimagery and topography: Comparing photogrammetric models processed with direct geo-referencing and ground control points. Geomatica. doi:10.5623/cig2016-102

Hugenholtz, C.H., Whitehead, K., Brown, O.W., Barchyn, T.E., Moorman, B.J., LeClair, A., Riddell, K., Hamilton, T., 2013. Geomorphological mapping with a small unmanned aircraft system (sUAS): Feature detection and accuracy assessment of a photogrammetrically-derived digital terrain model. Geomorphology 194, 16–24. doi:10.1016/j.geomorph.2013.03.023

Immerzeel, W.W., Kraaijenbrink, P.D.A., Shea, J.M., Shrestha, A.B., Pellicciotti, F., Bierkens, M.F.P., De Jong, S.M., 2014. High-resolution monitoring of Himalayan glacier dynamics using unmanned aerial vehicles. Remote Sens. Environ. 150, 93–103. doi:10.1016/j.rse.2014.04.025

James, M.R., Robson, S., 2014. Mitigating systematic error in topographic models derived from UAV and ground-based image networks. Earth Surf. Process. Landforms 39, 1413–1420. doi:10.1002/esp.3609

James, M.R., Robson, S., 2012. Straightforward reconstruction of 3D surfaces and topography with a camera: Accuracy and geoscience application. J. Geophys. Res. Earth Surf. 117, 1–17. doi:10.1029/2011JF002289

James, M.R., Robson, S., d'Oleire-Oltmanns, S., Niethammer, U., 2017. Optimising UAV topographic surveys processed with structure-from-motion: Ground control quality, quantity and bundle adjustment. Geomorphology 280, 51–66. doi:10.1016/J.GEOMORPH.2016.11.021

Kraaijenbrink, P., Meijer, S.W., Shea, J.M., Pellicciotti, F., De Jong, S.M., Immerzeel, W.W., 2016. Seasonal surface velocities of a Himalayan glacier derived by automated correlation of unmanned aerial vehicle imagery. Ann. Glaciol. 57, 103–113. doi:10.3189/2016AoG71A072

Tonkin, T.N., Midgley, N.G., 2016. Ground-control networks for image based surface reconstruction: An investigation of optimum survey designs using UAV derived imagery and structure-from-motion photogrammetry. Remote Sens. 8, 16–19. doi:10.3390/rs8090786

Tonkin, T.N., Midgley, N.G., Graham, D.J., Labadz, J.C., 2014. The potential of small unmanned aircraft systems and structure-from-motion for topographic surveys: A test of emerging integrated approaches at Cwm Idwal, North Wales. Geomorphology 226, 35–43. doi:10.1016/j.geomorph.2014.07.021

Westoby, M.J., Brasington, J., Glasser, N.F., Hambrey, M.J., Reynolds, J.M., 2012. "Structure-from-Motion" photogrammetry: A low-cost, effective tool for geoscience applications. Geomorphology 179, 300–314. doi:10.1016/j.geomorph.2012.08.021

Whitehead, K., Moorman, B.J., Hugenholtz, C.H., 2013. Brief Communication: Low-cost, on-demand aerial photogrammetry for glaciological measurement. Cryosphere 7, 1879–1884. doi:10.5194/tc-7-1879-2013

Wigmore, O., Mark, B., 2017. Monitoring tropical debris-covered glacier dynamics from high-resolution unmanned aerial vehicle photogrammetry, Cordillera Blanca, Peru. Cryosphere 11. doi:10.5194/tc-11-2463-2017

---

## Author Comment (AC1) · 14 Feb 2019

Dear Editor and Reviewers,

Thank you for your detailed comments and suggestions about our manuscript entitled "A high frequency and high resolution image time series of the Gornergletscher – Swiss Alps – derived from repeated UAV surveys".

We identified two main caveats noted by both reviewers. We answer hereafter to these concerns in a general manner with propositions of improvement, and then give detailed responses (with additional tests and quality checks when needed) to all comments in an itemized manner.

Response to the main concerns:

*Absence of actual GCPs.*

The first concern is related to the absence of actual GCPs to geo-reference our dataset, and to avoid internal deformations within the dataset.

We agree that the absence of GCPs can lead to some deformations in the master bundle adjustment (9th June), in particular a doming effect due to an imperfect lens calibration. However, while these deformations can lead to significant absolute positioning errors, we do not think that they translate in large errors for relative measurements, and in particular for the computed ice velocities. Indeed, velocities depend on the relative positioning errors, which we argue are small here (see response to reviewer#2 for details). This is because: (1) the internal deformations are relatively small (max. few meters) thanks to a careful photogrammetric processing, (2) these deformations are smoothly distributed in space thanks to the bundle adjustment of each campaign, and (3) the deformations are almost constant in time thanks to the co-registration procedure. It follows that for most of the expected uses of the present dataset, which focus on change detection, the errors imputable to the internal deformations of the master bundle adjustment can be regarded as negligible compared to other sources of uncertainty. As an illustration, the error in horizontal velocities over a two-weeks period that can be imputable to the internal deformations is less than 1mm/day (see response to reviewer#2 for details), which we believe is acceptable.

Having said this, we acknowledge that these points were not discussed enough in the manuscript. Furthermore, internal deformations must be carefully described, assessed and documented in the description of the dataset. We therefore propose the following modifications of our manuscript to address the reviewers' comments:

- Deformation errors will be described in detail in a new paragraph at the end of section 3.1: generation of co-registered orthomosaics and DEMs. We will include the references suggested by the reviewers, and inform the reader about the impact of internal deformations on our dataset.

- In addition, we adopted solution 3 proposed by Reviewer#2 to assess the internal deformations that may exist in the master bundle adjustment (June 9th). We therefore co-registered our master bundle adjustment on an orthomosaic provided by the Swiss mapping agency, SwissTopo. This orthomosaic has a 50 cm resolution and has been acquired in 2009. This led to two interesting products, which will be added to the dataset and discussed in the revised manuscript: (1) a transform function allowing to geo-reference our dataset in the Swiss national reference frame (CH1903-LV03), and (2) a map of residuals between adjusted and observed tie-point coordinates allowing to assess and visualize the internal deformations of the master bundle adjustment.

*Non-standard procedure to compute the Matching Maps.*

The second concern addressed by both reviewers is the application of a non-standard procedure to compute the matching maps.

We argue that we did follow a best-practice procedure but adapted it to match the requirements of the present application (see response to reviewer#1 for details). We tracked within-scene motions by block matching, which is the basis of the vast majority of change detection software used for ice velocity tracking (e.g. CIAS, ImCorr, COSI-Corr, GoLive). Compared to these tools, our feature tracking strategy only differs by some technical details, mostly driven by: (1) the need to handle high resolution data acquired by UAV (10 cm resolution) in contrast to the relatively coarse satellite images (few meters resolution) for which the aforementioned software have been designed; and (2) the wish to match every pixel of the input ortho-mosaics in order to generate dense matching maps and not sparse velocity fields. To ensure that the outputs of our Matching Map Maker utility are robust, we carried out some additional tests, which show that the velocities computed by our utility are in line with the ones obtained by two well-established glacier surface tracking algorithms (see response to reverer#1 for the detailed results).

This said, we fully understand the reviewers when they ask for better assessment and description of our feature tracking approach. We therefore propose the following modifications of our manuscript to answer these requests:
- We will describe in more detail the procedure we used to compute the matching maps, with particular attention paid to put our approach in context with existing methods (using the references suggested by both reviewers). To motivate these choices, we will add a paragraph focusing on the specific requirements of UAV acquisitions, the differences compared to satellite applications, and their implications for feature tracking.
- Building on a comment of Reviewer#1, we will discuss the sensitivity of the estimated velocity fields in relation to the similarity score used in the matching procedure.
- We will add the results of the benchmarking tests mentioned above to a sub-repository of the Matching Map Maker utility, and mention the results of these tests in the revised version of the manuscript. We hope this will help the users to evaluate the performance of our utility with respect to other tools.

Responses to the comments of Reviewer #1:

In the following point-by-point reply, RC denotes a reviewer comment and AR denotes our response to the comment.

RC: The first concern is based on the relative geo-referencing, the authors have chosen not to use real ground control points. While they have chosen one common bundle-adjustment as master, to "stitch" other control against it. It is common in photogrammetry to distribute GCP's and have them placed especially at the outer ends. This reduces "banana" bending, caused by imperfections in the lens model. However, this is not done in this study, thus one can assume such effects are here at hand as well. I am well aware of the logistics within such terrain, thus I am not asking to do this procedure. Nevertheless, I propose the authors put a bit more emphasis into describing the potential errors associated with this effect/shortcoming. I hope if this is done rightfully, it will reduce the mistake of over interpretation by other users of this data.

AR: We agree that internal deformations (e.g. banana bending) are possible, and we share the vision of Reviewer#1 that it is difficult to remediate to such deformations by measuring GCPs due to logistical constraints in high mountains.

We therefore completely approve the proposition to better describe potential errors, their amplitude, and their effect on the final products (i.e. orthomosaics, DEMs and matching maps) in order to inform the reader about the limitations of our dataset, and avoid over-interpretation of the possible uses.

To implement this idea, we propose to first discuss the internal deformations based on the literature in a new paragraph added at the end of section 'section 3.1: generation of co-registered orthomosaics and DEMs'. We will also discuss the impact of these internal deformations on the possible uses of our dataset.

To complement this theoretical error analysis, we also assessed the amplitude of the internal deformations in the master bundle adjustment of 9$^{th}$ June. To this end, we followed a proposition of Reviewer#2 and co-registered our master ortho-image on an orthomosaic produced by the Swiss mapping agency SwissTopo (and therefore properly referenced in the Swiss national reference frame). This orthomosaic has a 50 cm resolution and has been acquired in 2009. The residuals after co-registration provide information on the potential deformations within our master bundle adjustment (maximum few meters, see results of the co-registration in the answer to Reviewer#2). Thanks to this additional quality check, we will also be able to discuss the impact of the actual deformations of our dataset. We are aware that this procedure only assesses the horizontal deformations and that the vertical component can be more affected by the doming effects than the horizontal one, but for now we do not have access to a precise enough DEM to carry out a similar assessment on the vertical component. This will also be discussed in the new paragraph about internal deformations.

RC: My second concern is focused around the matching maps, which are less standard products, and therefore implementation details need to be discussed more. Although not standard, there do exist best-practices, and because the authors deviate from this I highlight some steps which may need more clarification, or adjustments, in order to improve the resulting matching map product or get a better understanding why certain steps are taken.

AR: We agree that matching maps must be better explained, with a particular focus on the procedure adopted to compute them. To this end, we will rewrite and improve the section '3.2 Surface displacement tracking: generation of Matching Maps' following three guidelines:
- Clearly state that Matching Maps are nothing but displacement maps.
- Contextualize our procedure with respect to well established software, highlighting similarities and differences.
- Benchmark our Matching Map Maker utility with respect to two well-established tools dedicated to displacement calculation by image correlation (we chose CIAS - https://www.mn.uio.no/geo/english/research/projects/icemass/cias/ and Imcorr - https://nsidc.org/data/velmap/imcorr.html as benchmarks). The results of the tests will be added to a sub-repository of the MMM source code, and will be mentioned in the main manuscript.

RC: The similarity score is "maximum absolute error" on grayscale images. This is a peculiar choice as throughout the season and throughout the day the sun has changing illumination directions. Though this similarity measure is very sensitive towards such effects. Commonly, the normalized cross correlation is used […].

AR: The similarity score we use is the minimum (not maximum) of the mean absolute difference between two patches. This similarity score is indeed uncommon for glacier image matching, but is

widely used in other domains and in particular for video tracking (e.g. Liu and Zaccarin, 1993; Chuang et al, 2014) because it is fast to compute, especially on large images using convolution. We therefore choose this score to accelerate the processing of our very large matching maps.

This said, we agree that this similarity score is in theory sensitive to illumination differences. However, in practice, we did not notice much adverse effects in shadow areas. This is mostly because the images were acquired roughly at the same time of the day (between 11:30 and 16:00), and because the orthoimages used to generate the matching maps are always separated by less than one month, which should mitigate the illumination differences.

To be sure that our choice of similarity score does not degrade the resulting matching maps, we tested two other scores in addition to the mean absolute differences, namely: (1) the normalized cross-correlation and (2) a high-pass filter (cutting wavelength=25m) coupled with a cross correlation (similar to (Fahnestock et al, 2016)). The test is carried out for the period July 13$^{th}$ – July 26$^{th}$ and focuses on a small area to limit the computing time. The results below (Fig. R1.1) show that the straight use of the normalized cross-correlation produces the less reliable results with many outliers (in white in the figure R1.1). In contrast, the cross correlation tuned for feature tracking in glacier context (i.e. with a preliminary filtering) and the mean absolute differences (the selected option for this manuscript) perform almost perfectly in the test area. In addition, one can notice that these two methods generate almost indistinguishable displacement maps, which tends to show that using mean absolute differences as similarity score results in robust displacement estimates.

[Figure]

*Figure R1.1: Surface displacements (July 13$^{th}$ – July 26$^{th}$) estimated by feature tracking using different similarity scores. (a) Situation map. (b) Displacements estimated by maximum normalized cross-correlation. (c) Displacements estimated by high pass filtering (cutting wavelength = 25 m) and maximum cross-correlation. (d) Displacements estimated by minimum of the mean absolute differences. In (b-d), all parameters of the tracking algorithm besides the similarity score are the same for the three scenarios (search template 201 x 201 pixels, search window 400 x 400 pixels).*

To make the reader aware of our choice of similarity score, we will expand the section '3.2 Surface displacement tracking: generation of Matching Maps' and mention the potential sensitivity of the Matching Map Maker utility to illumination differences. In addition, we will add the results of the tests shown above in a sub-repository of the Matching Map Maker utility and mention it in the manuscript. Finally, the MMM code in the repository has been updated to allow users to choose the similarity score used in feature tracking (the scores available are now: (1) normalized cross-correlation, (2) high pass filter + cross-correlation, and (3) mean absolute differences).

References:

Chuang M-C., Hwang J-N., Williams K., Towler R.: Tracking Live Fish from Low-Contrast and Low-Frame-Rate Stereo Videos, IEEE Transactions on Circuits and Systems for Video Technology, 25(1), 167-179, 2015.

Liu B., Zaccarin, A.: New Fast Algorithms for the Estimation of Block Motion Vectors, IEEE Transactions on Circuits and Systems for Video Technology, 3(2), 148-157, 1993.
Fahnestock M., Scambos T., Moon T., Gardner A., Haran T., Klinger M., Remote Sensing of Environment, 185, 84-94, 2016.

RC: […] the images are either pre-filtered with a high-pass-filter [Fahnestock 2016] or a Wallis-filter [Dehecq 2018]. I expect this will improve the results considerable.

AR: A first reason to pre-filter the images is the improvement of the normalization factor when the cross-correlation is used as similarity score. This is clearly visible in the results of the previous test assessing different similarity scores (Fig R1.1). But when mean absolute differences are used (as is the case in our manuscript), no normalization is required, and there is therefore no need for pre-filtering to help normalizing the similarity score.

Another argument for using pre-filtering is to focus on features with a characteristic scale close to the resolution of the images. We think that such use of pre-filtering is mostly relevant in case of matching satellite images with meter-scale resolution or coarser. In our case (10cm resolution), a high-pass-filter with a cutting wavelength close to the resolution of the images will extract very small features such as small superficial cracks, shadows or snow patches, which are probably not very persistent in time and thus not relevant to track. Instead, we prefer to use the full information of the patch of interest (5m x 5m) that also includes larger and more persistent features.

RC: Also the use of orientation-correlation [Heid 2012] or COSI-CORR [leprince 2007] might be a more robust procedure.

AR: To ensure that our Matching Map Maker utility is reliable, we benchmarked it by comparison with velocity maps calculated using the well-established CIAS and Imcorr programs. The results below (Fig. R1.2 and R1.3) show that our method does not introduce any bias or inconsistencies.

[Figure]

*Figure R1.3: Comparison of estimated displacements derived from the Matching map (top plots) and from the CIAS software (bottom plots). Period of interest: July 13th – July 26th. (a) Northward displacement, (b) Eastward displacement. The only noticeable differences are due to residual noise in the CIAS results due to the simple set up we applied.*

[Figure]

*Figure R1.2: Comparison of estimated displacements derived from the Matching map (top plots) and from the ImCorr software (bottom plots). Period of interest: July 13ᵗʰ – July 26ᵗʰ. (a) Northward displacement, (b) Eastward displacement. The only noticeable differences are due to residual noise in the ImCorr results due to the simple set up we applied.*

To reassure the user about the reliability of our utility, we will add the results above in the repository of the Matching Map Maker utility. In addition, we will mention the existence of this test (and the associated results) in the manuscript.

RC: The matching maps seem to have an integer displacement, and no sub pixel localization is applied. While these procedures are able to increase the precision considerably.

AR: We think that sub-pixel localization is mostly useful when the amplitude of the displacement between two dates is small with regard to the resolution of the images. This is the case for instance when glacier velocity is derived from satellite images, or when satellite images are used to measure ground deformations induced by tectonics (typical application of COSI-corr). However, in our case, the resolution of the ortho-images used to compute the matching maps is high (10 cm) compared to the expected displacement of the glacier (20-50 cm/day => 2.8-7 m displacement in two weeks). We therefore think that the absence of sub-pixel localization in our procedure is not a major issue.

RC: The spacing of velocity product is at pixel level, though in products like GoLIVE for example, the spacing of the grid is as large as the template size (in this case 300 meters). Similar processing in SAR speckle tracking is done with only 50% overlap [van wychen 2018]. This is done to have independent measurements, but now there will be a large smearing effect. It is possible to get to the resolution of a pixel, when pyramidal matching is applied, up to a point where optical flow can to be implemented.

AR: Yes the spacing of the Matching Maps is at the pixel level, and we think this is an advantage of this product rather than a drawback. Indeed, it allows the user to relate directly any pixel of a given orthomosaic to its counterpart in the following orthomosaic, without resorting to an interpolation of the velocity map. This makes the Matching Maps a convenient tool to navigate between the different acquisition dates.
But it is true that this also generates two drawbacks during the processing of the Matching Maps: (1) the matching procedure has to be applied to each pixel of the orthomosaic, which requires high computation. That is why we choose to boost the matching procedure by using the minimum of the

mean absolute difference as similarity score. (2) The results of the matching procedure are correlated for neighboring pixels. But a similar effect will appear if we compute sparse matching maps and then interpolate the results. And the advantage of the dense correlation is that the smoothness we introduce is not uniform. If there are large velocity contrasts, e.g. in a shear zone, this will be visible in our Matching Maps. In contrast, a sparse correlation coupled with an interpolation will probably miss it.

However, we agree that it is worth mentioning these aspects in the manuscript. We will therefore discuss these points in more details in section 3.2.

RC – minor comments.
AR: All minor comments will be implemented in the revised version of the manuscript. Hereafter we respond in details to three minor comments which we consider require a detailed answer.

RC: I assume all flights are nadir, or did the UAV also took oblique imagery. This is of interest, as it enhance the separation between internal parameters [james 2014].

AR: Due to the UAV platform we used (ebee from SenseFly), all flights are (unfortunately for lens calibration) nadir flights. We will specify it more clearly in section '2.2 Data acquisition' and discuss the implication on lens calibration in section '3.1 Generation of co-registered orthomosaics and DEMs'.

RC: "GCPs" is maybe not the correct term, as they are not real ground control, hence (manual) tie-points might be more correct.

AR: We fully agree with this terminology and we will adopt it in the revised version of the manuscript. We will also emphasize more the relative nature of our referencing.

RC: here seem to be multiple flights per campaign. Because a fixed wing is used, the landing must have had an impact. Hence, internal camera parameters might have been different between different flights. Thus, are these groups of data also separated in the camera optimization?

AR: Indeed, there are multiple flights per campaign, and only one set of internal parameters per campaign. Hence no separation of each group of data in the camera optimization. Since the SODA camera we used is specially designed for ebee flights (and associated possible rough landings) we supposed that the camera is robust to landing conditions, and therefore we expected that the camera parameters do not change much between flights.

To verify it, we plotted the camera parameters estimated for each date. In case of changing camera parameters induced by landing, different dates should have rather different estimated parameters. This is fortunately not the case. The estimated camera parameters are pretty stable along time (see figure R1.4 below), except one big 'jump' between July 13[th] and July 26[th]. This jump is due to the change of the camera, the device used for the first five acquisition dates having been destroyed in a UAV crash between these two dates during another (independent) project. It does not affect our data because the camera was never changed between the flights used for a single orthomosaic. In conclusion, we believe that the camera parameters are pretty stable along time, and are not much affected by the intra-campaign landings within a single day.

[Figure]

*Figure R1.4: temporal variability of estimated internal camera parameters. Vertical red lines denote a change of camera (between 07/13 and 07/26).*

In the following point-by-point reply, EC denotes an editor comment and AR denotes our response to the comment.

EC: I understand that you have used a commercial software package for the bulk part of the processing. This is ok, however, I believe the underlying algorithms should be explained/mentioned explicitly so that your study remains reproducible. For example, for people without access to pix4D Mapper, it is not helpful to state that the "default values" (p5/l5) were chosen.

AR: We agree with this comment, and we will therefore better explain the structure from motion (SfM) algorithms implemented in Pix4D, with references not only to the use of this software but also more general references explaining the use of SfM for glacier mapping.

EC: Fig. 5 has peculiar gray box behind that image which should be removed (maybed this is PDF viewer dependent? I have used Preview on MacOS.

AR : We will improve this figure.

EC: Consider changing the chosen colormaps to a sequential colormap. The rainbow colormaps (and their relatives) are prone to introduce false boundaries (e.g. https://www.climate-lab-book.ac.uk/2016/why-rainbow-colour-scales-can-be-misleading/ by Ed Hawkins).

AR: We will use a sequential colormap in our figures.

EC: Fig. 1c: Label Zwilings-GL is hard to read. Maybe move it to the heaven-part of the picture and use arrows? Not sure.

AR: We will improve this figure.

---

## Author Comment (AC2) · 14 Feb 2019

Dear Editor and Reviewers,

Thank you for your detailed comments and suggestions about our manuscript entitled "A high frequency and high resolution image time series of the Gornergletscher – Swiss Alps – derived from repeated UAV surveys".

We identified two main caveats noted by both reviewers. We answer hereafter to these concerns in a general manner with propositions of improvement, and then give detailed responses (with additional tests and quality checks when needed) to all comments in an itemized manner.

Response to the main concerns:

**Absence of actual GCPs.**

The first concern is related to the absence of actual GCPs to geo-reference our dataset, and to avoid internal deformations within the dataset.

We agree that the absence of GCPs can lead to some deformations in the master bundle adjustment (9$^{th}$ June), in particular a doming effect due to an imperfect lens calibration. However, while these deformations can lead to significant absolute positioning errors, we do not think that they translate in large errors for relative measurements, and in particular for the computed ice velocities. Indeed, velocities depend on the relative positioning errors, which we argue are small here (see response to reviewer#2 for details). This is because: (1) the internal deformations are relatively small (max. few meters) thanks to a careful photogrammetric processing, (2) these deformations are smoothly distributed in space thanks to the bundle adjustment of each campaign, and (3) the deformations are almost constant in time thanks to the co-registration procedure. It follows that for most of the expected uses of the present dataset, which focus on change detection, the errors imputable to the internal deformations of the master bundle adjustment can be regarded as negligible compared to other sources of uncertainty. As an illustration, the error in horizontal velocities over a two-weeks period that can be imputable to the internal deformations is less than 1mm/day (see response to reviewer#2 for details), which we believe is acceptable.

Having said this, we acknowledge that these points were not discussed enough in the manuscript. Furthermore, internal deformations must be carefully described, assessed and documented in the description of the dataset. We therefore propose the following modifications of our manuscript to address the reviewers' comments:

- Deformation errors will be described in detail in a new paragraph at the end of section 3.1: generation of co-registered orthomosaics and DEMs. We will include the references suggested by the reviewers, and inform the reader about the impact of internal deformations on our dataset.

- In addition, we adopted solution 3 proposed by Reviewer#2 to assess the internal deformations that may exist in the master bundle adjustment (June 9th). We therefore co-registered our master bundle adjustment on an orthomosaic provided by the Swiss mapping agency, SwissTopo. This orthomosaic has a 50 cm resolution and has been acquired in 2009. This led to two interesting products, which will be added to the dataset and discussed in the revised manuscript: (1) a transform function allowing to geo-reference our dataset in the Swiss national reference frame (CH1903-LV03), and (2) a map of residuals between adjusted and observed tie-point coordinates allowing to assess and visualize the internal deformations of the master bundle adjustment.

**Non-standard procedure to compute the Matching Maps.**

The second concern addressed by both reviewers is the application of a non-standard procedure to compute the matching maps.

We argue that we did follow a best-practice procedure but adapted it to match the requirements of the present application (see response to reviewer#1 for details). We tracked within-scene motions by block matching, which is the basis of the vast majority of change detection software used for ice velocity tracking (e.g. CIAS, ImCorr, COSI-Corr, GoLive). Compared to these tools, our feature tracking strategy only differs by some technical details, mostly driven by: (1) the need to handle high resolution data acquired by UAV (10 cm resolution) in contrast to the relatively coarse satellite images (few meters resolution) for which the aforementioned software have been designed; and (2) the wish to match every pixel of the input ortho-mosaics in order to generate dense matching maps and not sparse velocity fields. To ensure that the outputs of our Matching Map Maker utility are robust, we carried out some additional tests, which show that the velocities computed by our utility are in line with the ones obtained by two well-established glacier surface tracking algorithms (see response to revewer#1 for the detailed results).

This said, we fully understand the reviewers when they ask for better assessment and description of our feature tracking approach. We therefore propose the following modifications of our manuscript to answer these requests:
- We will describe in more detail the procedure we used to compute the matching maps, with particular attention paid to put our approach in context with existing methods (using the references suggested by both reviewers). To motivate these choices, we will add a paragraph focusing on the specific requirements of UAV acquisitions, the differences compared to satellite applications, and their implications for feature tracking.
- Building on a comment of Reviewer#1, we will discuss the sensitivity of the estimated velocity fields in relation to the similarity score used in the matching procedure.
- We will add the results of the benchmarking tests mentioned above to a sub-repository of the Matching Map Maker utility, and mention the results of these tests in the revised version of the manuscript. We hope this will help the users to evaluate the performance of our utility with respect to other tools.

Responses to the comments of Reviewer #2:

In the following point-by-point reply, RC denotes a reviewer comment and AR denotes our response to the comment.

RC: In this case it appears the authors used the later method relying on consumer grade GPS positions of the UAV to generate the 9 June orthomosaic and DEM, all other datasets are then coregistered to this 'master' dataset using stable bedrock features. What this means is that the stacks themselves are co-registered accurately and we can therefore observe changes between images pairs fairly precisely. However, the master 9 June dataset is likely to be so inaccurate positionally that any measurements made from these data are likely to be highly inaccurate.

AR: We agree with the fact that our dataset is not accurately geo-referenced, and we will remediate to this problem by applying the solution (3) proposed by Reviewer#2, and update our manuscript accordingly. We also agree that some internal deformations are possible in the master bundle adjustment (June 9th), in particular due to uncertainties in lens calibration. However, we rather disagree with the statement 'any measurements made from these data are likely to be highly inaccurate'. Indeed, since our dataset is self-consistent thanks to a careful co-registration, we think

that most of relative measurements (and in particular change measurements) are precise, despite the absence of absolute geo-referencing. To better explain this point, we discuss hereafter the impact of possible internal deformations on our dataset, and illustrate it in the case of computing velocity maps from the dataset (i.e. we discuss the fraction of the error in the Matching Maps that can be imputed to the internal deformations, and show that it is small).

As expected by both reviewers from a theoretical error analysis, and verified in practice by our additional tests (see figure R2.1 below), internal deformations are limited to a metric amplitude and are smooth in space (i.e. spread across the whole area, and not concentrated at a single location). Therefore, balanced by the size of the study area (few kilometers), the relative error induced by the deformations is on the order of 1/1000. Roughly speaking, one can expect a 1m error when measuring 1km wide objects, or a 10 cm error (i.e. close to the resolution of the final products) when measuring 100m wide features. But one should keep in mind that thanks to the co-registration procedure, the same deformations apply to the whole dataset (i.e. to each acquisition date), which mitigates dramatically their impact in all change detection applications. For instance, in case of horizontal velocity measurements (i.e. what is reported in our matching maps), the order of magnitude of glacier displacement in two weeks is 30 cm/day x 14 days = 4.2 m. Since deformations are similar for both dates (due to the co-registration procedure), the error related to internal deformations is a scaling factor of 1/1000 on distance measurement, i.e. in this particular case: 4.2m x 1/1000=4.2 mm. It follows that the error in velocity due to internal deformations is 4.2mm/14days=0.3 mm/day, which is in our opinion negligible compared to other error sources.

All in all, we believe that because changes are of limited amplitude (e.g. ice ablation reaches few cm/day, and ice flow <1m/day in the Gornergletscher ablation zone), the errors generated by internal deformations are small compared to other sources of uncertainty for most of change detection applications. In consequence, we believe that our dataset can be safely used to measure relative changes over time, with a decent accuracy (although this is not the case for absolute positions).

RC: Below I have outlined a few potential methods for remedying this issue. 1) Install ground control targets in the survey area and conduct another survey of the glacier, then use this as the master dataset to co-register the other datasets to, using stable/no change locations. This is that was done in the (Immerzeel et al., 2014; Kraaijenbrink et al., 2016) articles that the authors use as justification for the co-registration method they employed. Obviously installing ground control targets on glaciers and over a large area is extremely difficult and often not possible given access, safety and logisitical challenges. This is one of the major limitations of collecting research grade UAV datasets over challenging environments. Because of these limitations we are often forced to live with poorly distributed or inadequate numbers of ground targets. In the glacier context this means ground control is often restricted to installations along the moraine edges, but even this is much better than none. And, would in my opinion provide an acceptable product in line with other UAV/glacier publications. Furthermore, the authors did use some 'on-ice' GNSS data to assess the accuracy of the velocity measurements - these stations could be used in conjunction with moraine ground control targets alongside perhaps two more on ice targets - e.g. one near terminus, one near middle, limiting labour/time investment. 2) Fly a UAV equipped with L1 or L1/L2 differential GNSS and use an RTK or PPK positional solution to derive a more accurate master data set. Then use this dataset to co-register the other dates. An example of this method as applied to glaciers is currently under review in the cryosphere discussions: Chudley et al., 2018. 3) If there is a high quality DEM and imagery data from other sources e.g. LiDAR/airborne imagery/high resolution satellite data, then authors could extract ground control positions from these data and co-register the UAV datasets to this.

AR: We agree with the need of (1) assessing the internal deformations in the master bundle adjustment, and (2) better geo-referencing our dataset in the legal reference system of Switzerland

(CH1903 – LV03), especially for potential users that may require absolute positioning. To this end, we implemented the proposition 3) of Reviewer#2, and co-registered our master bundle adjustment on a 50 cm-resolution orthomosaic provided by the Swiss national mapping agency (SwissTopo). In the revised version of the manuscript, the results (Fig. R2.1) will be used to improve the section '4.1 Bundle adjustment and co-registration' following two guidelines:

- Provide to the user the transformation parameters allowing to reference our dataset (currently in imprecise WGS84 – UTM32) in the Swiss national reference frame (CH1903 – LV03).

- Assess the magnitude of the internal deformations of the master bundle adjustment by analyzing the residuals on the tie points used for the co-registration. Figure R2.1 shows that the internal deformations are mild (max few meters in the horizontal component) and spread over the whole area of interest. We are aware that this procedure only assesses the horizontal deformations and that the vertical component can be more affected by the doming effects than the horizontal one, but for now we do not have access to a precise enough DEM to carry out a similar assessment on the vertical component. This will also be discussed in the revised version of the manuscript.

[Figure]

*Figure R2.1: Residuals after co-registration of the master bundle adjustment on a 50 cm-resolution and accurately referenced orthoimage acquired in 2009 and provided by the Swiss national mapping agency. The transformation used for co-registration is a 2D-similarity. The resulting residuals can be interpreted as a proxy of the internal deformations of the master bundle adjustment (June 9$^{th}$ 2017).*

RC: Finally, I have read the other posted review for this paper and agree with their suggestions regarding the matching maps. Further description of how these were derived is needed as they are non-standard, and ideally more widely used methods for velocity field derivation should be applied – e.g. COSI-CORR as this would be 1) easier to follow, 2) likely to be more useful to other researchers.

AR: We agree that matching maps require more explanations, and that a thoughtful comparison with more widely used methods is needed. We therefore compared our Matching Maps with velocity maps computed with two well-established correlation algorithms (CIAS and ImCorr), as well as with velocity fields computed with our utility but with different similarity scores (e.g. normalized cross correlation). The results of these tests show that our procedure is robust and produces reliable Matching Maps (see response to Reviewer#1 for details).

To capitalize on these positive results, we will improve the section '3.2 Surface displacement tracking: generation of Matching Maps' of our manuscript following three guidelines (as proposed in the response to Reviewer#1 comments):

- Clearly state that Matching Maps are nothing but displacement maps.
- Contextualize our procedure with respect to well established software, with a clear explanation of which aspects are similar and which are different.
- Add the results of the benchmark of our Matching Map Maker utility with respect to CIAS and ImCorr to the repository of the MMM source code, and mention these results in the main manuscript.

RC: Additional points: To prevent the use of incorrect data by others the authors should clip out obviously erroneous regions in the DEM and orthomosaic, i.e. along the edges of the survey area. Ideally all datasets could be clipped to a common extent boundary and raster grid for ease of use, though this is less important.

AR: We agree with these points, and we will therefore add quality masks for the orthomosaics to our dataset (as was already done for the Matching Maps).

Responses to the comments of the Editor:

In the following point-by-point reply, EC denotes an editor comment and AR denotes our response to the comment.

EC: I understand that you have used a commercial software package for the bulk part of the processing. This is ok, however, I believe the underlying algorithms should be explained/mentioned explicitly so that your study remains reproducible. For example, for people without access to pix4D Mapper, it is not helpful to state that the "default values" (p5/l5) were chosen.

AR: We agree with this comment, and we will therefore better explain the structure from motion (SfM) algorithms implemented in Pix4D, with references not only to the use of this software but also more general references explaining the use of SfM for glacier mapping.

EC: Fig. 5 has peculiar gray box behind that image which should be removed (maybed this is PDF viewer dependent? I have used Preview on MacOS.

AR : We will improve this figure.

EC: Consider changing the chosen colormaps to a sequential colormap. The rainbow colormaps (and their relatives) are prone to introduce false boundaries (e.g. https://www.climate-lab-book.ac.uk/2016/why-rainbow-colour-scales-can-be-misleading/ by Ed Hawkins).

AR: We will use a sequential colormap in our figures.

EC: Fig. 1c: Label Zwilings-GL is hard to read. Maybe move it to the heaven-part of the picture and use arrows? Not sure.

AR: We will improve this figure.

---

## Editor Comment (EC1) · Reinhard Drews (Editor) · 15 Feb 2019

Dear Authors,

thank you for your response to the constructive and helpful concerns raised by the reviewers. I encourage submission of a revised manuscript (and please also submit a track-changes version). The paper will then go into re-review.

Please note that the paper can only be published if all concerns are satisfactorily addressed, and if the revised manuscript meets the standards of ESSD.

Kind regards, Reinhard Drews

---

## Author Response (AR1)

Dear Editor and Reviewers,

Thank you for your detailed comments and suggestions about our manuscript entitled "A high frequency and high resolution image time series of the Gornergletscher – Swiss Alps – derived from repeated UAV surveys".

The manuscript has been revised accordingly. Please find hereafter an item-by-item response to your comments.

Hoping that our responses answer your concerns, and that our propositions of improvements will fulfil your expectations,

Best regards,

Lionel Benoit, on behalf of all co-authors.
* * *
Responses to the comments of Reviewer #1:

In the following point-by-point reply, RC denotes a reviewer comment and AR denotes our response to the comment.

RC: The first concern is based on the relative geo-referencing, the authors have chosen not to use real ground control points. While they have chosen one common bundle-adjustment as master, to "stitch" other control against it. It is common in photogrammetry to distribute GCP's and have them placed especially at the outer ends. This reduces "banana" bending, caused by imperfections in the lens model. However, this is not done in this study, thus one can assume such effects are here at hand as well. I am well aware of the logistics within such terrain, thus I am not asking to do this procedure. Nevertheless, I propose the authors put a bit more emphasis into describing the potential errors associated with this effect/shortcoming. I hope if this is done rightfully, it will reduce the mistake of over interpretation by other users of this data.

AR: The impact of the absence of GCPs during the master bundle adjustment is now discussed in a new paragraph at the end of the section 3.1 'generation of co-registered orthomosaics and DEMs' (p6 – l 1-20). In this new paragraph we mention the possible existence of internal deformations and we describe in detail the impact of the absence of GCPs on geo-referencing.
In addition, the actual amplitude of the internal deformations is now assessed by studying the residuals of the co-registration of the orthomosaic derived from the master bundle adjustment with respect to an orthomosaic produced by the Swiss mapping agency. The amplitude of these deformations is shown to be small, and discussed in details at the end of the section 4.1 dedicated to the quality assessment of the orthomosaics (p 9, l15-22 -> p10, l 1-7). In addition, the residuals of the co-registration procedure are now displayed in the revised Fig 4.

RC: My second concern is focused around the matching maps, which are less standard products, and therefore implementation details need to be discussed more. Although not standard, there do exist best-practices, and because the authors deviate from this I highlight some steps which may need more clarification, or adjustments, in order to improve the resulting matching map product or get a better understanding why certain steps are taken.

AR: To better explain the generation of the Matching Maps, we fully revised the section '3.2: Surface displacement tracking: generation of Matching Maps' (in particular p 6, l 20 -> p 7, l 20). When re-writing this section, we followed three guidelines:

- Clearly state that Matching Maps are nothing but displacement maps.
- Contextualize our procedure with respect to well established software, highlighting similarities and differences.
- Benchmark our Matching Map Maker utility with two well-established tools dedicated to displacement calculation by image correlation (we chose CIAS - https://www.mn.uio.no/geo/english/research/projects/icemass/cias/ and Imcorr - https://nsidc.org/data/velmap/imcorr.html as benchmarks). The results of these tests have been added to a sub-repository of the MMM source code, which is mentioned in the main manuscript (P7, L14-18).

RC: The similarity score is "maximum absolute error" on grayscale images.  This is a peculiar choice as throughout the season and throughout the day the sun has changing illumination directions.  Though this similarity measure is very sensitive towards such effects. Commonly, the normalized cross correlation is used […].

AR: The choice of MAE as similarity score is now motivated in the manuscript and contextualized with references (p7, l3-5). In addition, the impact of illumination effects on MAE is now discussed in details (p7, l5-9).

To ensure that the use of MAE as similarity score does not impair the results of the matching procedure, we compared MAE with other similarity scores, namely: (1) the normalized cross-correlation and (2) a high-pass filter (cutting wavelength=25m) coupled with a cross correlation (similar to (Fahnestock et al, 2016)). The results below, also reported in a sub-repository of the MMM source code and mentioned in the manuscript (p13, l11-12), show that MAE performs as good as a high-pass filter (cutting wavelength=25m) coupled with a cross-correlation, and outperforms the normalized cross-correlation.

[Figure]

*Figure: Surface displacements (July 13th – July 26th) estimated by feature tracking using different similarity scores. (a) Situation map. (b) Displacements estimated by maximum normalized cross-correlation. (c) Displacements estimated by high pass filtering (cutting wavelength = 25 m) and maximum cross-correlation. (d) Displacements estimated by minimum of the mean absolute differences.  In (b-d), all parameters of the tracking algorithm besides the similarity score are the same for the three scenarios (search template 21 x 21 pixels, search window 400 x 400 pixels).*

RC: […] the images are either pre-filtered with a high-pass-filter [Fahnestock 2016] or a Wallis-filter [Dehecq 2018]. I expect this will improve the results considerable.

AR: A first reason to pre-filter the images is the improvement of the normalization factor when the cross-correlation is used as similarity score. This is clearly visible in the results of the previous test assessing different similarity scores. But when mean absolute differences are used (as is the case in our manuscript), no normalization is required, and there is therefore no need for pre-filtering to help normalizing the similarity score.

Another argument for using pre-filtering is to focus on features with a characteristic scale close to the resolution of the images. We think that such use of pre-filtering is mostly relevant in case of matching satellite images with meter-scale resolution or coarser. In our case (10 cm resolution) a high-pass-filter with a cutting wavelength close to the resolution of the images extracts very small features such as small superficial cracks, shadows or snow patches, which are probably not very persistent in time and thus not relevant to track. Instead, we prefer to use the full information of the patch of interest (5m x 5m) that also includes larger and more persistent features.

RC: Also the use of orientation-correlation [Heid 2012] or COSI-CORR [leprince 2007] might be a more robust procedure.

AR: To ensure that our Matching Map Maker utility is reliable, we benchmarked it by comparison with velocity maps calculated using the well-established CIAS and Imcorr algorithms. The results below show that our method does not introduce bias nor inconsistencies. To reassure the user about the reliability of our utility, we added the figures below in a sub-directory of the repository of the Matching Map Maker utility. In addition, we mention the existence of these tests (and the associated results) in the manuscript (p7, l16-18 and p13, l7-13).

[Figure]

*Figure: Comparison of estimated displacements derived from the Matching map (top plots) and from the CIAS software (bottom plots). Period of interest: July 13th – July 26th. (a) Northward displacement, (b) Eastward displacement. The only noticeable differences are due to residual noise in the CIAS results due to the simple set up we applied.*

[Figure]

*Figure: Comparison of estimated displacements derived from the Matching map (top plots) and from the ImCorr software (bottom plots). Period of interest: July 13th – July 26th. (a) Northward displacement, (b) Eastward displacement. The only noticeable differences are due to residual noise in the ImCorr results due to the simple set up we applied.*

RC: The matching maps seem to have an integer displacement, and no sub pixel localization is applied. While these procedures are able to increase the precision considerably.

AR: We think that sub-pixel localization is mostly useful when the amplitude of the displacement between two dates is small with regard to the resolution of the images. This is the case for instance when glacier velocity is derived from satellite images, or when satellite images are used to measure ground deformations induced by tectonics (typical application of COSI-corr). However, in our case, the resolution of the ortho-images used to compute the matching maps is high (10cm) compared to the expected displacement of the glacier (20-50cm/day => 2.8-7m displacement in two weeks). We therefore think that the absence of sub-pixel localization in our procedure is not a major issue.

RC: The spacing of velocity product is at pixel level, though in products like GoLIVE for example, the spacing of the grid is as large as the template size (in this case 300 meters). Similar processing in SAR speckle tracking is done with only 50% overlap [van wychen 2018]. This is done to have independent measurements, but now there will be a large smearing effect. It is possible to get to the resolution of a pixel, when pyramidal matching is applied, up to a point where optical flow can to be implemented.

AR: Yes the spacing of the Matching Maps is at the pixel level, and we think this is an advantage of this product rather than a drawback. Indeed, it allows the user to relate directly any pixel of a given orthomosaic to its counterpart in the following orthomosaic, without resorting to an interpolation of the velocity map. This makes the Matching Maps a convenient tool to navigate between the different acquisition dates. In the revised version of the manuscript, this is now clearly stated p 6, l25-27.

Minor comments.

RC: p1 l1 high-frequency -> inter-seasonal; p1 l1 image -> elevation and velocity

AR: For now we decided to not change the title of the paper. Indeed, we think that inter-seasonal can be misleading because almost only summer months are covered, and we prefer to emphasize more on the raw data (i.e. UAV based images) than on the processing (elevation and velocity).

RC: p1 l13 new dataset -> new topographic dataset

AR: Ok.

RC: p1 l13 intensive: subjective text, I am aware this is hard work but not necessary information for the abstract

AR: Ok. We removed this text.

RC: p1 l14 summer 2017 -> summer of 2017
    p2 l7 summer 2017 -> summer of 2017

AR: Ok.

RC: p1 l17 displacements and velocity, choose one as these are two words for the samething.
    p2 l17 displacements and velocities; choose one

AR: Ok. We chose 'displacements'.

RC: p2 l8 -> "10 consecutive DEMs and associated orthomosaics"; you first need to make a DEM in order to be able to make a mosaic

AR: This is right. We modified the manuscript accordingly (p 2, l7)

RC: p2 l9 surface evolution.

AR: Ok.

RC: p2 l23 "ice flow dynamics at the surface" I think the emergence velocity by [Brun 2018] is also a good example of a process. Furthermore, mentioning the use of multitemporalDEM's as in [Wang , Berthier ] might be worth it as well.

AR: Thank you for drawing our attention on these references. They are now used to better contextualize our work at the end of the introduction (p 2, l17-25).

RC: l1 "6%" maybe also specify in degrees

AR: Ok.

RC: p3 l4 "mainstream" -> central flowline?

AR: Ok.

RC: p3 l19 maybe include the lake location, which drains, and also include the ELA?

AR: The lake was moraine-dammed and disappeared some years ago due to glacier retreat. Regarding the ELA we think that it is roughly visible in the background orthomosaic in Fig 1b which has been acquired at the end of the summer.

RC: p5 l3 pix4D, please give the version

AR: Ok.

RC: p5 l7 swap orthomosaic and DEMs

AR: Ok.

RC: p5 l9 UTM_zone32 -> Universal Transverse Mercator (UTM zone 32)

AR: Ok. In addition, we better explain the georeferencing of the master bundle adjustment (p 6, l3-4).

RC: p5 l12 I assume all flights are nadir, or did the UAV also took oblique imagery. This isof interest, as it enhance the separation between internal parameters [james 2014].

AR: Yes all flights are close to nadir. In the revised version of the manuscript we specify it p 4, l4-5.

RC: p6 l1 "GCPs" is maybe not the correct term, as they are not real ground control, hence (manual) tie-points might be more correct.

AR: We agree with 'manual tie points'. We adopted this terminology throughout the revised manuscript.

RC: p6 l10 51x51 pixels, also include the metric size, that is roughly 5 meters right?
    p6 l12 2000 pixels, see former comment

AR: Ok.

RC: p7 l10 "locations with strong spatial gradients", this is not a very effective post-processing step. It is very local and isolated inliers will also be removed. More advanced post-processing techniques are possible [maksymiuk 2016], but I am not asking to do this, just so you are aware of such studies.

AR: We agree that our post-processing approach is rather simple. We therefore clearly state it in the revised manuscript (p 7, l24-26). In addition we added the reference suggested by Reviewer#1.

RC: p7 l16 "are interpolated", which type/method?

AR: We used a bilinear interpolation. This is now mentioned p 7, l34.

RC: p7 l16 "reliable measurements", not correct wording, there is no real testing, thus reliable is misplaced here.

AR: We reformulated this sentence (p7, l34) and changed 'reliable' to 'non-masked'.

RC: p8 l1 the east and northward components do not seem to be the same....? also, why is the displacement given in pixel and not in a metric scale?

AR: We did not understand the first part of the comment. Maybe a color issue? In any case Fig 3 has been redesigned according to a comment of the Editor. We hope that this change helps.
Regarding the second part of the comment, displacements are given in pixels because the unit of the Matching Maps is pixel. Displacements in m/day are shown on Figure 5.

RC: p8 l14 There seem to be multiple flights per campaign. Because a fixed wing is used, the landing must have had an impact. Hence, internal camera parameters might been different between different flights. Thus, are these groups of data also separated in the camera optimization?

AR: Yes there are multiple flights per campaign, and they are processed all together. This is now clearly stated p 5, l3.
Since the SODA camera we used is specially designed for ebee flights (and associated possible rough landings) it is supposed to be robust to landing conditions. Hence, the camera parameters are expected to not change much between flights.
To verify it, we plotted the camera parameters estimated for each date. In case of changing camera parameters induced by landing, different dates should have rather different estimated parameters. This is fortunately not the case. The estimated camera parameters are pretty stable along time (see figure R1.4 below), except one big 'jump' between July 13$^{th}$ and July 26$^{th}$. This jump is due to the change of the camera, the device used for the first five acquisition dates having been destroyed in a UAV crash between these two dates during another (independent) project. It does not affect our data because the camera was never changed between the flights used for a single orthomosaic. In conclusion, we believe that the camera parameters are pretty stable along time, and are not much affected by the intra-campaign landings within a single day.

[Figure]

*Figure R1.4: temporal variability of estimated internal camera parameters. Vertical red lines denote a change of camera (between 07/13 and 07/26).*

RC: p9 l3 are due -> can be

AR: Ok.

RC: p10 l11 "smooth out local variability", why is this done? and why is the mean taken, and not more robust measures like the median?

AR: Just to remove possible imperfections. But the formulation of this sentence was misleading. We removed this part of the sentence.

RC: p11 l12 why are single points used as validation, it might be good to look at stable terrain as well. The Randolph glacier mask can exclude glacial terrain, the rest can be used to create a histogram of displacements.

AR: Points located in stable terrain are also used for validation. We emphasize it in the revised manuscript (p 11, l 4). These are the green dots (and associated velocities) in Fig 5. The apparent displacements of these supposedly static locations are therefore used to estimate the precision of the velocities derived from the Matching Maps, and reported in Fig 5b.

RC: p11 l12 "since the area of interest", is only the icefall of interest? Are other features on the glacier tongue of interest as well (meander evolution of supra glacial streams, emergence velocity ....)?

AR: This is right, the icefall is not the only area of interest. This sentence has been modified to avoid misinterpretation (p 12, l1).

RC: p13 l4 "for almost any glacier mapping task", not specific and does not hold either, please rephrase.

AR: Ok.

Responses to the comments of Reviewer #2:

In the following point-by-point reply, RC denotes a reviewer comment and AR denotes our response to the comment.

RC: In this case it appears the authors used the later method relying on consumer grade GPS positions of the UAV to generate the 9 June orthomosaic and DEM, all other datasets are then coregistered to this 'master' dataset using stable bedrock features. What this means is that the stacks themselves are co-registered accurately and we can therefore observe changes between images pairs fairly precisely. However, the master 9 June dataset is likely to be so inaccurate positionally that any measurements made from these data are likely to be highly inaccurate.

AR: We agree with the fact that based on consumer grade GPS data, the initial geo-referencing of our dataset is not precise. To warn the reader about it, a new paragraph has been added at the end of the section 3.1 (generation of co-registered orthomosaics and DEMs, p 6, l1-13). This paragraph details the impact of the absence of GCPs on the geo-referencing of the master bundle adjustment, and mentions that internal deformations can also occur. To better geo-reference our dataset, we followed the option (3) proposed by reviewer#2. We therefore co-registered the master orthomosaic on an orthomosaic processed by the Swiss Mapping Agency. The parameters of the transformation allowing to link our dataset to the Swiss reference system are given in Table 2 of the revised manuscript.
Regarding the internal deformations, they are now assessed by studying the residuals of aforementioned co-registration procedure. The amplitude of these deformations is discussed in details at the end of the section 4.1 dedicated to the quality assessment of the orthomosaics (p 9, l15-22 -> p10, l 1-7). In addition, the residuals of the co-registration procedure are displayed in the revised Fig 4 and are relatively small. Furthermore, we show that the influence of these deformations has a very small impact on the estimated velocities (see our replies to the next comment).

RC: Below I have outlined a few potential methods for remedying this issue. 1) Install ground control targets in the survey area and conduct another survey of the glacier, then use this as the master dataset to co-register the other datasets to, using stable/no change locations. This is that was done in the (Immerzeel et al., 2014; Kraaijenbrink et al., 2016) articles that the authors use as justification for the co-registration method they employed. Obviously installing ground control targets on glaciers and over a large area is extremely difficult and often not possible given access, safety and logisitical challenges. This is one of the major limitations of collecting research grade UAV datasets over challenging environments. Because of these limitations we are often forced to live with poorly distributed or inadequate numbers of ground targets. In the glacier context this means ground control is often restricted to installations along the moraine edges, but even this is much better than none. And, would in my opinion provide an acceptable product in line with other UAV/glacier publications. Furthermore, the authors did use some 'on-ice' GNSS data to assess the accuracy of the velocity measurements - these stations could be used in conjunction with moraine ground control targets alongside perhaps two more on ice targets - e.g. one near terminus, one near middle, limiting labour/time investment. 2) Fly a UAV equipped with L1 or L1/L2 differential GNSS and use an RTK or PPK positional solution to derive a more accurate master data set. Then use this dataset to co-register the other dates. An example of this method as applied to glaciers is currently under review in the cryosphere discussions: Chudley et al., 2018. 3) If there is a high quality DEM and imagery data from other sources e.g. LiDAR/airborne imagery/high resolution satellite data, then authors could extract ground control positions from these data and co-register the UAV datasets to this.

AR: As mentioned above, we adopted the solution (3) proposed by Reviewer#2 to better geo-reference our dataset and to assess the amplitude of the internal deformations. The revised Fig 4 shows that the internal deformations have a metric amplitude and are smoothly spread over the target area. Taking into account the size of the area of interest, the relative errors induced by the internal deformations are therefore of the order of 1/1000. When propagating in Matching Maps, these internal deformations lead to errors <1mm/day on the estimated surface velocity, which is very small in regard to the amplitude of the ice surface velocity signal (around 30cm/day). To warn the reader about the potential impact of the internal deformations on our dataset, a comprehensive uncertainty analysis is now reported in a new paragraph added at the end of section 4.1 (p9, l15 -> p10, l6).

RC: Finally, I have read the other posted review for this paper and agree with their suggestions regarding the matching maps. Further description of how these were derived is needed as they are non-standard, and ideally more widely used methods for velocity field derivation should be applied – e.g. COSI-CORR as this would be 1) easier to follow, 2) likely to be more useful to other researchers.

AR: As mentioned in our response to the comments of Reviewer#1, we completely revised the section dedicated to the generation of Matching Maps (in particular p 6, l 20 -> p 7, l 20) in order to better describe this product.
In addition, to ensure that our Matching Map Maker utility is reliable, we benchmarked against velocity maps calculated using the well-established CIAS and Imcorr algorithms. The results show that our method does not introduce any bias or inconsistencies. To reassure the user about the reliability of our utility, we added the results of the benchmarking tests in a sub-directory of the repository of the Matching Map Maker utility. In addition, we mention the existence of these tests (and the associated results) in the manuscript (p7, l16-18 and p13, l7-13).

RC: Additional points: To prevent the use of incorrect data by others the authors should clip out obviously erroneous regions in the DEM and orthomosaic, i.e. along the edges of the survey area.

Ideally all datasets could be clipped to a common extent boundary and raster grid for ease of use, though this is less important.

AR: We prefer to warn the user about the limitations of the dataset rather than clipping the 'best' areas. To this end, we provide in the dataset a processing report for each pair DEM + orthomosaic. These reports allow the user to identify, for each acquisition date, in which areas our products fulfills its needs for a given application. Regarding the Matching Maps, unreliable areas are masked, as highlighted in the revised version of Fig 3. The reliability masks are available along with the Matching Maps.
* * *
Responses to the comments of the Editor:

In the following point-by-point reply, EC denotes an editor comment and AR denotes our response to the comment.

EC: I understand that you have used a commercial software package for the bulk part of the processing. This is ok, however, I believe the underlying algorithms should be explained/mentioned explicitly so that your study remains reproducible. For example, for people without access to pix4D Mapper, it is not helpful to state that the "default values" (p5/l5) were chosen.

AR: We modified the first paragraph of section 3.1 to better explain the photogrammetric processing using Pix4D (p 5, l3-10).

EC: Fig. 5 has peculiar gray box behind that image which should be removed (maybed this is PDF viewer dependent? I have used Preview on MacOS.

AR: We modified figure 5 to make it more clear. To this end, the ice surface velocity is now displayed with contour lines rather than a background color. We hope that this change also improves the visual aspect of the figure.

EC: Consider changing the chosen colormaps to a sequential colormap. The rainbow colormaps (and their relatives) are prone to introduce false boundaries (e.g. https://www.climate-lab-book.ac.uk/2016/why-rainbow-colour-scales-can-be-misleading/ by Ed Hawkins).

AR: Fig 3 has been edited, and in the new version a grayscale colormap is used.

EC: Fig. 1c: Label Zwilings-GL is hard to read. Maybe move it to the heaven-part of the picture and use arrows? Not sure.

AR: Ok.

[revised manuscript text omitted]

---

## Author Response (AR2)

Dear Editor and Reviewer,

Thank you for your new comments about our manuscript entitled "A high resolution image time series of the Gornergletscher – Swiss Alps – derived from repeated UAV surveys".

The manuscript has been revised accordingly. In particular, the title has been changed to avoid misinterpretation of the term 'high frequency'. In addition, proper displacement maps at a 5m resolution and expressed in m/day are now available in addition to the full resolution Matching Maps. The manuscript has been edited to introduce these displacement maps, and the dataset has been updated accordingly.

Hoping that these changes answer your last concerns,

Best regards,

Lionel Benoit, on behalf of all co-authors.

[revised manuscript text omitted]